# Faster Stochastic Algorithms for Minimax Optimization under Polyak-Łojasiewicz Conditions

**Lesi Chen**
School of Data Science
Fudan University
lschen19@fudan.edu.cn

**Boyuan Yao**
School of Data Science
Fudan University
byyao19@fudan.edu.cn

**Luo Luo**[*]
School of Data Science
Fudan University
luoluo@fudan.edu.cn

## Abstract

This paper considers stochastic first-order algorithms for minimax optimization under Polyak-Łojasiewicz (PL) conditions. We propose SPIDER-GDA for solving the finite-sum problem of the form $\min_x \max_y f(x, y) \triangleq \frac{1}{n} \sum_{i=1}^n f_i(x, y)$, where the objective function $f(x, y)$ is $\mu_x$-PL in $x$ and $\mu_y$-PL in $y$; and each $f_i(x, y)$ is $L$-smooth. We prove SPIDER-GDA could find an $\epsilon$-approximate solution within $\mathcal{O}\left((n + \sqrt{n}\,\kappa_x\kappa_y^2)\log(1/\epsilon)\right)$ stochastic first-order oracle (SFO) complexity, which is better than the state-of-the-art method whose SFO upper bound is $\mathcal{O}\left((n + n^{2/3}\kappa_x\kappa_y^2)\log(1/\epsilon)\right)$, where $\kappa_x \triangleq L/\mu_x$ and $\kappa_y \triangleq L/\mu_y$. For the ill-conditioned case, we provide an accelerated algorithm to reduce the computational cost further. It achieves $\tilde{\mathcal{O}}\left((n + \sqrt{n}\,\kappa_x\kappa_y)\log^2(1/\epsilon)\right)$ SFO upper bound when $\kappa_y \gtrsim \sqrt{n}$. Our ideas also can be applied to the more general setting that the objective function only satisfies PL condition for one variable. Numerical experiments validate the superiority of proposed methods.

## 1 Introduction

This paper focuses on smooth minimax optimization problem of the form

$$\min_{x \in \mathbb{R}^{d_x}} \max_{y \in \mathbb{R}^{d_y}} f(x, y) \triangleq \frac{1}{n} \sum_{i=1}^n f_i(x, y), \tag{1}$$

which covers a lot of important applications in machine learning such as reinforcement learning [10, 42], AUC maximization [13, 24, 48], imitation learning [5, 32], robust optimization [11], causal inference [28], game theory [6, 29] and so on.

We are interested in the minimax problems under PL conditions [9, 32, 45]. The PL condition [35] was originally proposed to relax the strong convexity in minimization problem that is sufficient for achieving the global linear convergence rate for first-order methods. In machine learning community, it has been successfully used to analyze the convergence behavior for overparameterized neural networks [23], robust phase retrieval [40] and a plenty of fundamental models [18]. There are many popular minimax formulations only satisfy PL condition, but lack strong convexity (or strong concavity). The examples include PL-game [32], robust least square [45], deep AUC maximization [24] and generative adversarial imitation learning of LQR [5, 32].

Yang et al. [45] showed that the alternating gradient descent ascent (AGDA) algorithm linearly converges to the saddle point when the objective function satisfies two-sided PL condition. They also proposed the SVRG-AGDA method for the finite-sum problem (1), which could find $\epsilon$-approximate

---

[*]The corresponding author

36th Conference on Neural Information Processing Systems (NeurIPS 2022).

Table 1: We present the comparison of SFO complexities under two-sided PL condition. Note that Yang et al. [45] named their stochastic algorithm as variance-reduced-AGDA (VR-AGDA). Here we call it SVRG-AGDA to distinguish with other variance reduced algorithms.

| Algorithm | Complexity | Reference |
|---|---|---|
| GDA/AGDA | $\mathcal{O}\left(n\kappa_x\kappa_y^2\log\left(1/\epsilon\right)\right)$ | Theorem B.1, [45] |
| SVRG-AGDA | $\mathcal{O}\left((n+n^{2/3}\kappa_x\kappa_y^2)\log\left(1/\epsilon\right)\right)$ | [45] |
| SVRG-GDA | $\mathcal{O}\left((n+n^{2/3}\kappa_x\kappa_y^2)\log\left(1/\epsilon\right)\right)$ | Theorem C.1 |
| SPIDER-GDA | $\mathcal{O}\left((n+\sqrt{n}\kappa_x\kappa_y^2)\log\left(1/\epsilon\right)\right)$ | Theorem 4.1 |
| AccSPIDER-GDA | $\begin{cases}\tilde{\mathcal{O}}\left(\sqrt{n}\kappa_x\kappa_y\log^2\left(1/\epsilon\right)\right), & \sqrt{n}\lesssim\kappa_y; \\ \tilde{\mathcal{O}}\left(n\kappa_x\log^2\left(1/\epsilon\right)\right), & \kappa_y\lesssim\sqrt{n}\lesssim\kappa_x\kappa_y; \\ \mathcal{O}\left((n+\sqrt{n}\kappa_x\kappa_y^2)\log\left(1/\epsilon\right)\right), & \kappa_x\kappa_y\lesssim\sqrt{n}.\end{cases}$ | Theorem 5.1 |

solution within $\mathcal{O}\left((n+n^{2/3}\kappa_x\kappa_y^2)\log(1/\epsilon)\right)$ stochastic first-order oracle (SFO) calls,[2] where $\kappa_x$ and $\kappa_y$ are the condition numbers with respect to PL condition for $x$ and $y$ respectively. The variance reduced technique in the SVRG-AGDA leads to better a convergence rate than full batch AGDA whose SFO complexity is $\mathcal{O}(n\kappa_x\kappa_y^2\log(1/\epsilon))$. However, there are still some open questions left. Firstly, Yang et al. [45]'s theoretical analysis heavily relies on the alternating update rules. It remains interesting whether a simultaneous version of GDA (or its stochastic variants) also has similar convergence results. Secondly, it is unclear whether the SFO upper bound obtain by SVRG-AGDA can be improved by designing more efficient algorithms.

For one-sided PL condition, we desire to find the stationary point of $g(x) \triangleq \max_{y\in\mathbb{R}^{d_y}} f(x,y)$, since the saddle point may not exist. Nouiehed et al. [32] proposed the multi-step GDA method that achieves the $\epsilon$-stationary point within $\mathcal{O}(\kappa_y^2 L\epsilon^{-2}\log(\kappa_y/\epsilon))$ numbers of full gradient iterations. The similar complexity also can be obtained by AGDA [45]. Recently, Yang et al. [47] proposed the smoothed-AGDA that improves the upper bound into $\mathcal{O}(\kappa_y L\epsilon^{-2})$. Both multi-step GDA Nouiehed et al. [32] and smoothed-AGDA [46] can be extended to online setting [14], but the formulation (1) with finite-sum structure has not been explored.

In this paper, we introduce a variance reduced first-order method, called SPIDER-GDA, which constructs the gradient estimator by stochastic recursive gradient and the iterations are based on simultaneous gradient descent ascent. We prove that SPIDER-GDA could achieve $\epsilon$-approximate solution of the two-sided PL problem of the form (1) within $\mathcal{O}\left((n+\sqrt{n}\kappa_x\kappa_y^2)\log(1/\epsilon)\right)$ SFO calls, which has better dependency on $n$ than SVRG-AGDA [45]. We also provide an acceleration framework to improve first-order methods for solving ill-conditioned minimax problems under PL conditions. The accelerated SPIDER-GDA (AccSPIDER-GDA) could achieve $\epsilon$-approximate solution within $\tilde{\mathcal{O}}\left((n+\sqrt{n}\kappa_x\kappa_y)\log^2(1/\epsilon)\right)$ SFO calls[3] when $\kappa_y \gtrsim \sqrt{n}$, which is the best known SFO upper bound for this problem. We summarize our main results and compare them with related work in Table 1. Without loss of generality, we always suppose $\kappa_x \gtrsim \kappa_y$. Furthermore, the proposed algorithms also work for minimax problem with one-sided PL condition. We present the results for this case in Table 2.

## 2 Related Work

The minimax optimization problem (1) can be viewed as the following minimization problem

$$\min_{x\in\mathbb{R}^{d_x}} \left\{ g(x) \triangleq \max_{y\in\mathbb{R}^{d_y}} f(x,y) \right\}.$$

A natural way to solve such problem is the multi-step GDA algorithm [21, 25, 32, 36] that contains double-loop iterations in which the outer loop can be regarded as running inexact gradient descent on

---

[2]The original analysis [45] provided an SFO upper bound $\mathcal{O}\left((n+n^{2/3}\max\{\kappa_x^3,\kappa_y^3\})\log(1/\epsilon)\right)$, which can be refined to $\mathcal{O}\left((n+n^{2/3}\kappa_x\kappa_y^2)\log(1/\epsilon)\right)$ by some little modification in the proof.

[3]In this paper, we ues the notation $\tilde{\mathcal{O}}(\cdot)$ to hide the logarithmic factors of $\kappa_x, \kappa_y$ but not $1/\epsilon$.

Table 2: We present the comparison of SFO complexities under one-sided PL condition.

| Algorithm | Complexity | Reference |
|---|---|---|
| Multi-Step GDA | $\mathcal{O}(n\kappa_y^2 L\epsilon^{-2}\log(\kappa_y/\epsilon))$ | [32] |
| GDA/AGDA | $\mathcal{O}\left(n\kappa_y^2 L\epsilon^{-2}\right)$ | Theorem B.2, [45] |
| Smooothed-AGDA | $\mathcal{O}\left(n\kappa_y L\epsilon^{-2}\right)$ | [47] |
| SVRG-GDA | $\mathcal{O}\left(n + n^{2/3}\kappa_y^2 L\epsilon^{-2}\right)$ | Theorem F.1 |
| SPIDER-GDA | $\mathcal{O}\left(n + \sqrt{n}\kappa_y^2 L\epsilon^{-2}\right)$ | Theorem 6.1 |
| AccSPIDER-GDA | $\begin{cases} \mathcal{O}\left(\sqrt{n}\kappa_y L\epsilon^{-2}\log(\kappa_y/\epsilon)\right), & \sqrt{n} \lesssim \kappa_y; \\ \mathcal{O}\left(nL\epsilon^{-2}\log(\kappa_y/\epsilon)\right), & \kappa_y \lesssim \sqrt{n} \lesssim \kappa_y^2; \\ \mathcal{O}\left(n + \sqrt{n}\kappa_y^2 L\epsilon^{-2}\right), & \kappa_y^2 \lesssim \sqrt{n}. \end{cases}$ | Theorem 6.2 |

$g(x)$ and the inner loop finds the approximate solution to $\max_{y\in\mathbb{R}^{d_y}} f(x,y)$ for a given $x$. Another class of methods is the two-timescale (alternating) GDA algorithm [9, 21, 44, 45] that only has single-loop iterations which update two variables with different stepsizes. The two-timescale GDA method can be implemented more easily and typically performs better than multi-step GDA empirically. Its convergence rate also can be established by analyzing function $g(x)$ but the analysis is more challenging than the multi-step GDA.

The variance reduction is a popular technique to improve the efficiency of stochastic optimization algorithms [2–4, 7, 8, 12, 16, 17, 19, 27, 31, 33, 34, 37–39, 43, 49, 50]. It is shown that solving nonconvex minimization problems with stochastic recursive gradient estimator [12, 16, 34, 43, 51] has the optimal SFO complexity. In the context of minimax optimization, the variance reduced algorithms also obtain the best-known SFO complexities in several settings [1, 15, 25, 26, 41, 45]. Specifically, the (near) optimal SFO algorithm for several convex-concave minimax problem has been proposed [15, 26], but the optimality for the more general case is still unclear [25, 45].

The Catalyst acceleration [20] is a useful approach to reduce the computational cost of ill-conditioned optimization problems, which is based on a sequence of inexact proximal point iterations. Lin et al. [22] first introduced Catalyst into minimax optimization. Later, Luo et al. [26], Tominin et al. [41], Yang et al. [46] designed the accelerated stochastic algorithms for convex-concave and nonconvex-concave problem. Recently, Yang et al. [47] also applied this technique to one-sided PL setting.

## 3 Notation and Preliminaries

First of all, we present the definition of saddle point.

**Definition 3.1.** *We say $(x^*, y^*) \in \mathbb{R}^{d_x} \times \mathbb{R}^{d_y}$ is a saddle point of function $f : \mathbb{R}^{d_x} \times \mathbb{R}^{d_y} \to \mathbb{R}$ if it holds that $f(x^*, y) \leq f(x^*, y^*) \leq f(x, y^*)$ for any $x \in \mathbb{R}^{d_x}$ and $y \in \mathbb{R}^{d_y}$.*

Then we formally define the Polyak-Łojasiewicz (PL) condition [35] as follows.

**Definition 3.2.** *We say a differentiable function $h : \mathbb{R}^d \to \mathbb{R}$ satisfies $\mu$-PL for some $\mu > 0$ if $\|\nabla h(z)\|^2 \geq 2\mu\big(h(z) - \min_{z'\in\mathbb{R}^d} h(z')\big)$ holds for any $z \in \mathbb{R}^d$.*

Note that the PL condition does not require the strongly convexity and it can be satisfied even if the function is nonconvex [18].

We are interested in the finite-sum minimax optimization problem (1) under following assumptions.

**Assumption 3.1.** *We suppose each component $f_i : \mathbb{R}^{d_x} \times \mathbb{R}^{d_y} \to \mathbb{R}$ is $L$-smooth, i.e., there exists a constant $L > 0$ such that $\|\nabla f_i(x,y) - \nabla f_i(x',y')\|^2 \leq L^2\big(\|x - x'\|^2 + \|y - y'\|^2\big)$ holds for any $x, x' \in \mathbb{R}^{d_x}$ and $y, y' \in \mathbb{R}^{d_y}$.*

**Assumption 3.2.** *We suppose the differentiable function $f : \mathbb{R}^{d_x} \times \mathbb{R}^{d_y} \to \mathbb{R}$ satisfies two-sided PL condition, i.e., there exist constants $\mu_x > 0$ and $\mu_y > 0$ such that $f(\cdot, y)$ is $\mu_x$-PL for any $y \in \mathbb{R}^{d_y}$ and $-f(x, \cdot)$ is $\mu_y$-PL for any $x \in \mathbb{R}^{d_x}$.*

Under Assumption 3.1 and 3.2, we define the condition numbers of problem (1) with respect to PL conditions for $x$ and $y$ as $\kappa_x \triangleq L/\mu_x$ and $\kappa_y \triangleq L/\mu_y$ respectively.

We also introduce the following assumption for the existence of saddle point.

**Assumption 3.3** (Yang et al. [45]). *We suppose the function $f : \mathbb{R}^{d_x} \times \mathbb{R}^{d_y} \to \mathbb{R}$ has at least one saddle point $(x^*, y^*)$. We also suppose that for any fixed $y \in \mathbb{R}^{d_y}$, the problem $\min_{x \in \mathbb{R}^{d_x}} f(x, y)$ has a nonempty solution set and a finite optimal value; and for any fixed $x \in \mathbb{R}^{d_x}$, the problem $\max_{y \in \mathbb{R}^{d_y}} f(x, y)$ has a nonempty solution set and a finite optimal value.*

The goal of solving minimax problem under two-sided PL condition is finding an $\epsilon$-approximate solution or $\epsilon$-saddle point that is defined as follows.

**Definition 3.3.** *We say $x$ is an $\epsilon$-approximate solution of problem (1) if it holds that $g(x) - g(x^*) \leq \epsilon$, where $g(x) = \max_{y \in \mathbb{R}^{d_y}} f(x, y)$.*

**Definition 3.4.** *Under Assumption 3.3, we say $(x, y)$ is an $\epsilon$-saddle point of problem (1) if it holds that $\|x - x^*\|^2 + \|y - y^*\|^2 \leq \epsilon$ for some saddle point $(x^*, y^*)$.*

We allow the saddle point does not exist for the problem with one-sided PL condition. In such case, it is guaranteed that $g(x) \triangleq \max_{y \in \mathbb{R}^{d_y}} f(x, y)$ is differentiable [32, Lemma A.5] and we target to find an $\epsilon$-stationary point of $g(x)$.

**Definition 3.5.** *If the function $g : \mathbb{R}^{d_x} \to \mathbb{R}$ is differentiable, we say $x$ is an $\epsilon$-stationary point of $g$ if it holds that $\|\nabla g(x)\| \leq \epsilon$.*

## 4 A Faster Algorithm for the Two-Sided PL Condition

We first consider the two-sided PL conditioned minimax problem of the finite-sum form (1) under Assumption 3.1, 3.2 and 3.3. We propose a novel stochastic algorithm, which we refer to as SPIDER-GDA. The detailed procedure of our method is presented in Algorithm 1. SPIDER-GDA constructs the stochastic recursive gradient estimators [12, 31] as follows:

$$G_x(x_{t,k}, y_{t,k}) = \frac{1}{B} \sum_{i \in S_x} \left( \nabla_x f_i(x_{t,k}, y_{t,k}) - \nabla_x f_i(x_{t,k-1}, y_{t,k-1}) + G_x(x_{t,k-1}, y_{t,k-1}) \right),$$

$$G_y(x_{t,k}, y_{t,k}) = \frac{1}{B} \sum_{i \in S_y} \left( \nabla_y f_i(x_{t,k}, y_{t,k}) - \nabla_y f_i(x_{t,k-1}, y_{t,k-1}) + G_y(x_{t,k-1}, y_{t,k-1}) \right).$$

It simultaneously updates two variables $\mathbf{x}$ and $\mathbf{y}$ by estimators $G_x$ and $G_y$ with different stepsizes $\tau_x = \Theta(1/(\kappa_y^2 L))$ and $\tau_y = \Theta(1/L)$ respectively. Huang et al. [16], Luo et al. [25], Xian et al. [44] have studied the SPIDER-type algorithm for nonconvex-strongly-concave problem and showed it converges to the stationary point of $g(x) \triangleq \max_{y \in \mathbb{R}^{d_y}} f(x, y)$ sublinearly. However, solving the problem minimax problems with two-sided PL condition desires stronger linear convergence rate, which leads to our theoretical analysis be different from previous work.

We measure the convergence of SPIDER-GDA by the following Lyapunov function

$$\mathcal{V}_{t,k} \triangleq g(x_{t,k}) - g(x^*) + \frac{\lambda \tau_x}{\tau_y} \left( g(x_{t,k}) - f(x_{t,k}, y_{t,k}) \right),$$

where $x^* \in \arg\min_{x \in \mathbb{R}^{d_x}} g(x)$ and $\lambda = \Theta(\kappa_y^2)$. We can establish recursion for $\mathcal{V}_{t,k}$ as follows

$$\mathbb{E}[\mathcal{V}_{t,K}] \leq \mathbb{E}\left[ \mathcal{V}_{t,0} - \frac{\tau_x}{16} \left( 2 - \frac{M}{B} \right) \sum_{k=0}^{K-1} \|G_x(x_{t,k}, y_{t,k})\|^2 - \frac{\lambda \tau_x}{16} \left( 2 - \frac{M}{B} \right) \sum_{k=0}^{K-1} \|G_y(x_{t,k}, y_{t,k})\|^2 \right].$$

Using the above inequality by setting $M = B = \sqrt{n}$ leads to the estimators $G_x(\tilde{x}_t, \tilde{y}_t)$ and $G_y(\tilde{x}_t, \tilde{y}_t)$ be sufficiently close to the exact gradient and converge to zero linearly, which indicates $g(\tilde{x}_t)$ also converges to $g(x^*)$ linearly. We formally provide the convergence result for SPIDER-GDA in the following theorem and its detailed proof is shown in appendix.

**Theorem 4.1.** *Under Assumption 3.1, 3.2 and 3.3, we run Algorithm 1 with $M = B = \sqrt{n}$, $\tau_y = 1/(5L)$, $\lambda = 32L^2/\mu_y^2$, $\tau_x = \tau_y/(24\lambda)$, $K = \lceil 4224/(\mu_x \tau_x) \rceil$ and $T = \lceil \log(1/\epsilon) \rceil$. Then the output $(\tilde{x}_T, \tilde{y}_T)$ satisfies $g(\tilde{x}_T) - g(x^*) \leq \epsilon$ and $g(\tilde{x}_T) - f(\tilde{x}_T, \tilde{y}_T) \leq 24\epsilon$ in expectation; and it takes no more than $\mathcal{O}\big((n + \sqrt{n}\kappa_x \kappa_y^2) \log(1/\epsilon)\big)$ SFO calls.*

---

**Algorithm 1** SPIDER-GDA $(f, (x_0, y_0), T, K, M, B, \tau_x, \tau_y)$

---

1: $\tilde{x}_0 = x_0, \tilde{y}_t = y_0$

2: **for** $t = 0, 1, \ldots, T - 1$ **do**

3:     $x_{t,0} = \tilde{x}_t, y_{t,0} = \tilde{y}_t$

4:     **for** $k = 0, 1, \ldots, K - 1$ **do**

5:         **if** $\mod (k, M) = 0$ **then**

6:             $G_x(x_{t,k}, y_{t,k}) = \nabla_x f(x_{t,k}, y_{t,k})$

7:             $G_y(x_{t,k}, y_{t,k}) = \nabla_y f(x_{t,k}, y_{t,k})$

8:         **else**

9:             draw mini-batches $S_x$ and $S_y$ independently with both sizes of $B$.

10:             $G_x(x_{t,k}, y_{t,k}) = \frac{1}{B} \sum_{i \in S_x} [\nabla_x f_i(x_{t,k}, y_{t,k}) - \nabla_x f_i(x_{t,k-1}, y_{t,k-1}) + G_x(x_{t,k-1}, y_{t,k-1})]$

11:             $G_y(x_{t,k}, y_{t,k}) = \frac{1}{B} \sum_{i \in S_y} [\nabla_y f_i(x_{t,k}, y_{t,k}) - \nabla_y f_i(x_{t,k-1}, y_{t,k-1}) + G_y(x_{t,k-1}, y_{t,k-1})]$

12:         **end if**

13:         $x_{t,k+1} = x_{t,k} - \tau_x G_x(x_{t,k}, y_{t,k})$

14:         $y_{t,k+1} = x_{y,k} + \tau_y G_y(x_{t,k}, y_{t,k})$

15:     **end for**

16:     choose $(\tilde{x}_{t+1}, \tilde{y}_{t+1})$ from $\{(x_{t,k}, y_{t,k})\}_{k=0}^{K-1}$ uniformly at random.

17: **end for**

18: **return** $(\tilde{x}_T, \tilde{y}_T)$

---

**Algorithm 2** AccSPIDER-GDA

---

1: $u_0 = x_0$

2: **for** $k = 0, 1, \ldots, K - 1$ **do**

3:     $(x_{k+1}, y_{k+1}) = \text{SPIDER-GDA}\big(f(x, y) + \frac{\beta}{2}\|x - u_k\|^2, (x_k, y_k), T_k, K, M, B, \tau_x, \tau_y\big)$

4:     $u_{k+1} = x_{k+1} + \gamma(x_{k+1} - x_k)$

5: **end for**

6: **option I** (two-sided PL): **return** $(x_K, y_K)$

7: **option II** (one-sided PL): **return** $(\hat{x}, \hat{y})$ chosen uniformly at random from $\{(x_k, y_k)\}_{k=0}^{K-1}$

---

Our results provide an SFO upper bound of $\mathcal{O}((n + \sqrt{n}\kappa_x \kappa_y^2) \log(1/\epsilon))$ for finding an $\varepsilon$-approximate solution that is better than the complexity $\mathcal{O}((n + n^{2/3}\kappa_x \kappa_y^2) \log(1/\epsilon))$ derived from SVRG-AGDA [45]. It is possible to use SVRG-type [17, 49] estimators to replace the stochastic recursive estimators in Algorithm 1, which results the algorithm SVRG-GDA. We can prove that SVRG-GDA also has $\mathcal{O}((n + n^{2/3}\kappa_x \kappa_y^2) \log(1/\epsilon))$ SFO upper bound that matches the theoretical result of SVRG-AGDA. We provide the details in Appendix C.

## 5 Further Acceleration with Catalyst

Both the proposed SPIDER-GDA (Algorithm 1) and existing SVRG-AGDA [45] have the complexities more heavily depend on the condition number of $y$ than the condition number of $x$. It is natural to ask can we make the dependency of two condition numbers balanced like the results in the strongly-convex-strongly-concave case [22, 25, 41]. In this section, we show it is possible by introducing the Catalyst acceleration.

To make acceleration possible, we need to assume the uniqueness of the optimal set for inner problem.

**Assumption 5.1.** *We assume the inner problem* $\max_{y \in \mathbb{R}^{d_y}} f(x, y)$ *has an unique solution.*

We proposed the accelerated SPIDER-GDA (AccSPIDER-GDA) in Algorithm 2 for reducing the computational cost further. Each iteration of the algorithm solve the following sub-problem

$$\min_{x\in\mathbb{R}^{d_x}} \max_{y\in\mathbb{R}^{d_y}} F_k(x,y) \triangleq \min_{x\in\mathbb{R}^{d_x}} \left\{ g(x) + \frac{\beta}{2}\|x - u_k\|_2^2 \right\}. \tag{2}$$

by SPIDER-GDA (Algorithm 1). AccSPIDER-GDA has the following convergence result if the sub-problem attain the required accuracy.

**Lemma 5.1.** *Under Assumption 3.1, 3.2 and 3.3, we run Algorithm 2 by $\beta = 2L$, $\gamma = 0$ and the appropriate setting for the sub-problem solver such that $\mathbb{E}[\|x_k - \tilde{x}_k\|^2 + \|y_k - \tilde{y}_k\|^2] \leq \delta$, where $(\tilde{x}_k, \tilde{y}_k)$ is a saddle point of $F_{k-1}$ $(k \geq 1)$ and we set the precision*

$$\delta = \frac{\mu_x \epsilon}{11(\mu_x + 4L)L} \tag{3}$$

*Then it holds that*

$$\mathbb{E}[g(x_k) - g(x^*)] \leq \left(1 - \frac{\mu_x}{2\beta + \mu_x}\right)^k \left(g(x_0) - g(x^*)\right) + \frac{\epsilon}{2}.$$

The setting $\beta = \Theta(L)$ in Lemma 5.1 guarantees the sub-problem (2) has condition number of the order $\mathcal{O}(1)$ for $x$. It is more well-conditioned on $x$, we prefer to address the following equivalent problem

$$\max_{y\in\mathbb{R}^{d_y}} \min_{x\in\mathbb{R}^{d_x}} F_k(x,y) = - \min_{y\in\mathbb{R}^{d_y}} \max_{x\in\mathbb{R}^{d_x}} \left\{ -F_k(x,y) \right\}. \tag{4}$$

Since (4) is a minimax problem satisfying two sided PL condition, we can apply SPIDER-GDA to solve it. And we can show that under Assumption 5.1, the saddle point $(\tilde{x}_k, \tilde{y}_k)$ of each $F_{k-1}(k \geq 1)$ is unique (see Lemma E.2 in appendix) and we are able to obtain a good approximation to it.

**Lemma 5.2.** *Under Assumption 3.1, 3.2 and 3.3, if we use Algorithm 1 to solve each sub-problem $\max_{y\in\mathbb{R}^{d_y}} \min_{x\in\mathbb{R}^{d_x}} F_k(x,y)$ (2) with $\beta = 2L$, $M = B = \sqrt{n}$, $\tau_x = 1/(15L)$, $\lambda = 288$, $\tau_y = \tau_x/(24\lambda)$, $K = \lceil 4224/(\mu_y\tau_y)\rceil$, $T_k = \lceil\log(1/\delta_k)\rceil$, then it holds that*

$$\mathbb{E}[\|x_{k+1} - \tilde{x}_{k+1}\|^2 + \|y_{k+1} - \tilde{y}_{k+1}\|^2] \leq 7236\kappa_y^2 \delta_k \mathbb{E}[\|x_k - \tilde{x}_k\|^2 + \|y_k - \tilde{y}_k\|^2],$$

*where $(\tilde{x}_k, \tilde{y}_k)$ is the unique saddle point of $F_{k-1}(k \geq 1)$.*

For a short summary, Lemma 5.1 means Algorithm 2 requires $\mathcal{O}(\kappa_x \log(1/\epsilon))$ numbers of inexact proximal point iterations to find an $\epsilon$-approximate solution of the problem. And Lemma E.1 tells us that each sub-problem can be solved within a SFO complexity of $\mathcal{O}\left(n + \sqrt{n}\kappa_y\right)\log(1/\delta_k))$. Thus, the total complexity for AccSPIDER-GDA becomes $\mathcal{O}((n\kappa_x + \sqrt{n}\kappa_x\kappa_y)\log(1/\epsilon)\log(1/\delta_k))$. Our next step is to specify $\delta_k$ which would lead to the total SFO complexity of the algorithm.

**Theorem 5.1.** *Under Assumption 3.1, 3.2, 3.3 and 5.1 if we let $\gamma = 0, \beta = 2L$ and use Algorithm 1 to solve each sub-problem $\max_{y\in\mathbb{R}^{d_y}} \min_{x\in\mathbb{R}^{d_x}} F_k(x,y)$ (2) with $M, B, \tau_x, \tau_y, K$ defined as Lemma 5.2 and $T_k = \lceil\log(1/\delta_k)\rceil$, where*

$$\delta_k = \begin{cases} \frac{1}{7236\kappa_y^2} \min\left\{\frac{1}{4}, \frac{(\beta - L)\mu_y\delta}{16\beta^2\|x_k - x_{k-1}\|^2}\right\}, & k \geq 1; \\ \frac{\delta\mu_y}{14472\kappa_y^2(g(x_0) - g(x^*))}, & k = 0, \end{cases} \tag{5}$$

*and $\delta$ is followed by the definition in (3). Then Algorithm 2 can return $x_K$ such that $g(x_K) - g(x^*) \leq \epsilon$ in expectation with no more than $\mathcal{O}((n\kappa_x + \sqrt{n}\kappa_x\kappa_y)\log(1/\epsilon)\log(\kappa_x\kappa_y/\epsilon))$ SFO calls.*

Lemma 5.1 does not rely on the choice of sub-problem solver, we can apply the acceleration framework in Algorithm 2 by replacing SPIDER-GDA with other algorithms. We summarize the SFO complexities for the acceleration of different algorithms in Table 3.

## 6   Extension to One-Sided PL Condition

In this section, we show the idea that SPIDER-GDA and its Catalyst acceleration also work for one-sided PL condition. We relax Assumption 3.2 and 3.3 to the following one.

Table 3: Accelerated results for different methods under two-sided PL condition.

| Method | Before Acceleration | After Acceleration |
|--------|--------------------|--------------------|
| GDA | $\mathcal{O}(n\kappa_x\kappa_y^2 \log(1/\epsilon))$ | $\tilde{\mathcal{O}}\left(n\kappa_x\kappa_y \log^2(1/\epsilon)\right)$ |
| SVRG-GDA | $\mathcal{O}((n + n^{2/3}\kappa_x\kappa_y^2)\log(1/\epsilon))$ | $\begin{cases} \tilde{\mathcal{O}}\left(n^{2/3}\kappa_x\kappa_y \log^2(1/\epsilon)\right), & n^{1/3} \lesssim \kappa_y; \\ \tilde{\mathcal{O}}\left(n\kappa_x \log^2(1/\epsilon)\right), & \kappa_y \lesssim n^{1/3} \lesssim \kappa_x\kappa_y; \\ \text{no acceleration}, & \kappa_x\kappa_y \lesssim n^{1/3}. \end{cases}$ |
| SPIDER-GDA | $\mathcal{O}\left((n + \sqrt{n}\kappa_x\kappa_y^2)\log(1/\epsilon)\right)$ | $\begin{cases} \tilde{\mathcal{O}}\left(\sqrt{n}\kappa_x\kappa_y \log^2(1/\epsilon)\right), & \sqrt{n} \lesssim \kappa_y; \\ \tilde{\mathcal{O}}\left(n\kappa_x \log^2(1/\epsilon)\right), & \kappa_y \lesssim \sqrt{n} \lesssim \kappa_x\kappa_y; \\ \text{no acceleration}, & \kappa_x\kappa_y \lesssim \sqrt{n}. \end{cases}$ |

**Assumption 6.1.** *We suppose that* $-f(x, \cdot)$ *is* $\mu_y$-*PL for any* $x \in \mathbb{R}^{d_x}$; *the problem* $\max_{y\in\mathbb{R}^{d_y}} f(x, y)$ *has a nonempty solution set and an optimal value ;* $g(x) \triangleq \max_{y\in\mathbb{R}^{d_y}} f(x, y)$ *is lower bounded, i.e., we have* $g^* = \inf_{x\in\mathbb{R}^{d_x}} g(x) > -\infty$.

We first show that the SFO complexity of SPIDER-GDA outperforms SVRG-GDA [4] by a factor of $\mathcal{O}(n^{1/6})$ in Theorem 6.1.

**Theorem 6.1.** *Under Assumption 3.1 and 6.1 , Let* $T = 1$ *and* $M, B, \tau_x, \tau_y, \lambda$ *as defined in Theorem 4.1 and* $K = \lceil 64/(\tau_x\epsilon^2) \rceil$, *then Algorithm 1 can guarantee the output* $\hat{x}$ *to satisfy* $\|\nabla g(\hat{x})\| \leq \epsilon$ *in expectation with no more than* $\mathcal{O}(n + \sqrt{n}\kappa_y^2 L\epsilon^{-2})$ *SFO calls.*

The AccSPIDER-GDA also performs better than SPIDER-GDA in one-sided PL condition for ill conditioned case. In the following lemma, we show that AccSPIDER-GDA could find an approximate stationary point if we solve the sub-problem sufficiently accurate.

**Lemma 6.1.** *Under Assumption 3.1 and 6.1, if it holds true that* $\mathbb{E}[\|x_k - \tilde{x}_k\|^2 + \|y_k - \tilde{y}_k\|^2] \leq \delta$ *for some saddle point* $(\tilde{x}_k, \tilde{y}_k)$ *of* $F_{k-1}$ $(k \geq 1)$, *where*

$$\delta = \frac{\epsilon^2}{8L\kappa_y(22\mu_y + 1)}. \tag{6}$$

*Let* $\beta = 2L$, *then for the output* $(\hat{x}, \hat{y})$ *of Algorithm 2, it holds true that*

$$\mathbb{E}\|\nabla g(\hat{x})\|^2 \leq \frac{8\beta(g(x_0) - g^*)}{K} + \frac{\epsilon^2}{2}.$$

Compared with two-sided PL condition, the analysis of AccSPIDER-GDA is more complicated since the precision $\delta_k$ at each round are different. By choosing the parameters of the algorithm carefully, we obtain the following results.

**Theorem 6.2.** *Under Assumption 3.1, 6.1 and 5.1, if we run Algorithm 2 by* $\gamma = 0, \beta = 2L$ *and use Algorithm 1 to solve each sub-problem* $\max_{y\in\mathbb{R}^{d_y}} \min_{x\in\mathbb{R}^{d_x}} F_k(x, y)$ *(2) with* $M, B, \tau_x, \tau_y, \lambda, K$ *and* $T_k$ *(dependent on* $\delta$) *as in Theorem 5.1 and* $\delta$ *is followed by the definition in Lemma 6.1, then Algorithm 2 can find* $\hat{x}$ *such that* $\|\nabla g(\hat{x})\| \leq \epsilon$ *in expectation within* $\mathcal{O}((n + \sqrt{n}\kappa_y)L\epsilon^{-2}\log(\kappa_y/\epsilon))$ *SFO calls.*

We can directly set $\beta = 0$ for Algorithm 2 in the case of very large $n$ and in this case AccSPIDER-GDA reduces to SPIDER-GDA. The summary and comparison for the complexities for the one-sided PL condition is shown in Table 2. Besides, the algorithms of GDA and SVRG-GDA also can be accelerated with Catalyst framework and we present the corresponding results in Table 4.

---

[4]The complexity for finding an $\epsilon$-stationary point of SVRG-GDA in presented in Appendix F.

Table 4: Acceleration for different methods under one-sided PL condition.

| Method | Before Acceleration | After Acceleration | |
|---|---|---|---|
| GDA | $\mathcal{O}\left(n\kappa_y^2 L\epsilon^{-2}\right)$ | $\mathcal{O}\left(n\kappa_y L\epsilon^{-2}\log(\kappa_y/\epsilon)\right)$ | |
| SVRG-GDA | $\mathcal{O}\left(n + n^{2/3}\kappa_y^2 L\epsilon^{-2}\right)$ | $\begin{cases} \mathcal{O}\left(n^{2/3}\kappa_y L\epsilon^{-2}\log(\kappa_y/\epsilon)\right), & n^{1/3} \lesssim \kappa_y; \\ \mathcal{O}\left(nL\epsilon^{-2}\log(\kappa_y/\epsilon)\right), & \kappa_y \lesssim n^{1/3} \lesssim \kappa_y^2; \\ \text{no acceleration}, & \kappa_y^2 \lesssim n^{1/3}. \end{cases}$ | |
| SPIDER-GDA | $\mathcal{O}\left(n + \sqrt{n}\kappa_y^2 L\epsilon^{-2}\right)$ | $\begin{cases} \mathcal{O}\left(\sqrt{n}\kappa_y L\epsilon^{-2}\log(\kappa_y/\epsilon)\right), & \sqrt{n} \lesssim \kappa_y; \\ \mathcal{O}\left(nL\epsilon^{-2}\log(\kappa_y/\epsilon)\right), & \kappa_y \lesssim \sqrt{n} \lesssim \kappa_y^2; \\ \text{no acceleration}, & \kappa_y^2 \lesssim \sqrt{n}. \end{cases}$ | |

## 7 Experiments

In this section, we conduct the numerical experiments to show the advantage of proposed algorithms and the source code is available[5]. We consider the following two player Polyak-Łojasiewicz game:

$$\min_{x\in\mathbb{R}^d} \max_{y\in\mathbb{R}^d} f(x,y) \triangleq \frac{1}{2}x^\top Px - \frac{1}{2}y^\top Qy + x^\top Ry,$$

where

$$P = \frac{1}{n}\sum_{i=1}^n p_i p_i^\top, \quad Q = \frac{1}{n}\sum_{i=1}^n q_i q_i^\top \quad \text{and} \quad R = \frac{1}{n}\sum_{i=1}^n r_i r_i^\top.$$

We independently sample $p_i$, $q_i$ and $r_i$ from $\mathcal{N}(0, \Sigma_P)$, $\mathcal{N}(0, \Sigma_Q)$ and $\mathcal{N}(0, \Sigma_R)$ respectively. We set the covariance matrix $\Sigma_P$ as the form of $UDU^\top$ such that $U \in \mathbb{R}^{d\times r}$ is column orthogonal matrix and $D \in \mathbb{R}^{r\times r}$ is diagonal with $r < d$. The diagonal elements of $D$ are distributed uniformly in the interval $[\mu, L]$ with $0 < \mu < L$. The matrix $\Sigma_Q$ is set by the similar way to $\Sigma_P$. We also let $\Sigma_R = 0.1VV^\top$, where each element of $V \in \mathbb{R}^{d\times d}$ is sampled from $\mathcal{N}(0, 1)$ independently. Since the covariance matrices $\Sigma_P$ and $\Sigma_Q$ are rank-deficient, it is guaranteed that both $P$ and $Q$ are singular. Hence, the objective function is not strongly-convex and not strongly-concave, but it satisfies the two-sided PL-condition [18]. We set $n = 6000$, $d = 10$, $r = 5$, $L = 1$ for all experiments; and let $\mu$ be $10^{-5}$ and $10^{-9}$ for two different settings.

We compare the proposed SPIDER-GDA (Algorithm 1) and AccSPIDER-GDA (Algorithm 2) with the baseline algorithm SVRG-AGDA [45]. We let $B = 1$ and $M = n$ for all of these algorithms and both of the stepsizes for $x$ and $y$ are tuned from $\{10^{-1}, 10^{-2}, 10^{-3}, 10^{-4}, 10^{-5}\}$. For AccSPIDER, we set $\beta = L/(20n)$ and $\gamma = 0.999$. We present the results of the number of SFO calls against the norm of gradient and the distance to the saddle point in Figure 1 and Figure 2. It is clear that our algorithms outperform than baselines.

## 8 Conclusion and Future Work

In this paper, we have investigated stochastic optimization for PL conditioned minimax problem with the finite-sum objective. We have proposed the SPIDER-GDA algorithm, which reduces the dependency of the sample numbers in SFO complexity. Moreover, we have introduced a Catalyst scheme to accelerate our algorithm for solving the ill-conditioned problems. We improve the SFO upper bound of the state-of-the-art algorithms for both two-sided and one-sided PL conditions.

However, the optimality of SFO algorithms for the PL conditioned minimax problem is still unclear. It is interesting to construct the lower bound for verifying the tightness of our results. It is also possible to extend our algorithm to online setting.

---

[5] `https://github.com/TrueNobility303/SPIDER-GDA`

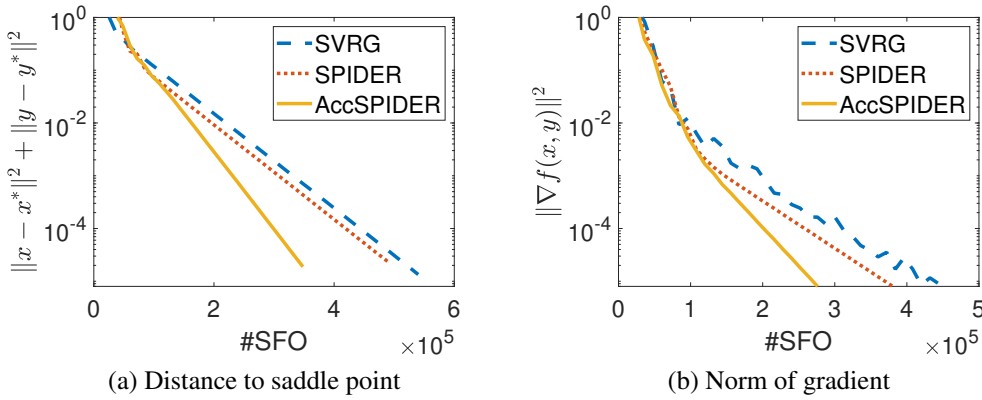

(a) Distance to saddle point         (b) Norm of gradient

Figure 1: The comparison for the case of $\mu = 10^{-5}$

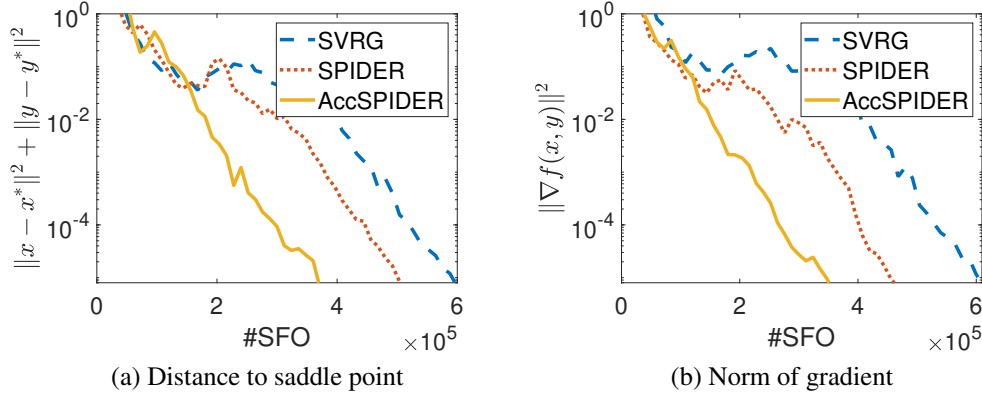

(a) Distance to saddle point         (b) Norm of gradient

Figure 2: The comparison for the case of $\mu = 10^{-9}$

## Acknowledgements

This work is supported by National Natural Science Foundation of China (No. 62206058) and Shanghai Sailing Program (22YF1402900).

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
