# A  Some Useful Lemmas

In this section, we provide some lemmas which are useful in the following proofs.

First of all, we define three notations of optimality.

**Definition A.1.** *We say $(x^*, y^*)$ is a saddle point of function $f$, if for all $(x, y)$, it holds that*

$$f(x^*, y) \leq f(x^*, y^*) \leq f(x, y^*).$$

*We say $(x^*, y^*)$ is a global minimax point, if for all $x \in \mathbb{R}^{d_x}, y \in \mathbb{R}^{d_y}$, it holds that*

$$f(x^*, y) \leq f(x^*, y^*) \leq \max_{y' \in \mathbb{R}^{d_y}} f(x, y').$$

*And we say $(x^*, y^*)$ is a stationary point, if it holds that*

$$\nabla_x f(x^*, y^*) = \nabla_y f(x^*, y^*) = 0.$$

For general nonconvex-nonconcave minimax problem, a stationary point or a global minimax point is weaker than a saddle point, i.e. a stationary point or a global minimax point may not be a saddle point. However, under two-sided PL condition, the above three notations are equivalent.

**Lemma A.1** (Yang et al. [45, Lemma 2.1]). *Under Assumption 3.2, it holds that*

$$(saddle\ point) \Leftrightarrow (global\ minimax\ point) \Leftrightarrow (stationary\ point).$$

*Further, if $(x^*, y^*)$ is a saddle point of $f$, then*

$$\max_{y \in \mathbb{R}^{d_y}} f(x^*, y) = f(x^*, y^*) = \min_{x \in \mathbb{R}^{d_x}} f(x, y^*).$$

*and vice versa.*

It is well known that weak duality always holds.

**Lemma A.2** (Nesterov [30, Theorem 1.3.1]). *Given a function $f$, we have*

$$\max_{y \in \mathbb{R}^{d_x}} \min_{x \in \mathbb{R}^{d_x}} f(x, y) \leq \min_{x \in \mathbb{R}^{d_x}} \max_{y \in \mathbb{R}^{d_y}} f(x, y).$$

It is a standard conclusion that the existence of saddle points implies strong duality. Since strong duality is important for the convergence of Catalyst scheme under PL condition, we present this lemma as follows.

**Lemma A.3.** *If $(x^*, y^*)$ is a saddle point of function $f$, then $(x^*, y^*)$ is also a global minimax point and stationary point of $f$, and it holds that*

$$\max_{y \in \mathbb{R}^{d_y}} \min_{x \in \mathbb{R}^{d_x}} f(x, y) = f(x^*, y^*) = \min_{x \in \mathbb{R}^{d_x}} \max_{y \in \mathbb{R}^{d_y}} f(x, y).$$

**Lemma A.4** (Yang et al. [45, Lemma A.1]). *Under Assumption 3.2, then $f(x, y)$ also satisfies the following quadratic growth condition, i.e. for all $x \in \mathbb{R}^{d_x}, y \in \mathbb{R}^{d_y}$, it holds that*

$$f(x, y) - \min_{x \in \mathbb{R}^{d_x}} f(x, y) \geq \frac{\mu_x}{2} \|x^*(y) - x\|^2,$$

$$\max_{y \in \mathbb{R}^{d_y}} f(x, y) - f(x, y) \geq \frac{\mu_y}{2} \|y^*(x) - y\|^2,$$

*where $x^*(y)$ is the projection of $y$ on the set $\arg\min_{x \in \mathbb{R}^{d_x}} f(x, y)$ and $y^*(x)$ is the projection of $x$ on the set of $\arg\max_{y \in \mathbb{R}^{d_y}} f(x, y)$.*

Also, we analyze the properties of function $g(x)$.

**Lemma A.5** (Yang et al. [45, Lemma 2.1]). *Under Assumption 3.2, then $g(x)$ satisfies $\mu_x$-PL, i.e. for all $x$ we have*

$$\|\nabla g(x)\|^2 \geq 2\mu_x(g(x) - g(x^*)).$$

**Lemma A.6** (Yang et al. [45, in the proof of Theorem 3.1]). *Under Assumption 3.2 and 3.1, then for all $x, y$ it holds true that*

$$\|\nabla_x f(x,y) - \nabla g(x)\|^2 \leq \frac{2L^2}{\mu_y}(g(x) - f(x,y)).$$

The above lemma is a direct result from quadratic growth property implied by PL condition and $L$-smooth property of function $f(x,y)$. Using the definition of $\mu_y$-PL in $y$, we can also show the relationship between $\|\nabla_x f(x,y) - \nabla g(x)\|^2$ and $\|\nabla_y f(x,y)\|^2$ as follows.

**Lemma A.7.** *Under Assumption 3.2 and 3.1, then for all $x, y$ it holds true that*

$$\|\nabla_x f(x,y) - \nabla g(x)\|^2 \leq \frac{L^2}{\mu_y^2}\|\nabla_y f(x,y)\|^2$$

**Lemma A.8** (Nouiehed et al. [32, Lemma A.5]). *Under Assumption 6.1 and 3.1, then $g(x)$ satisfies $(L + L^2/\mu_y)$-smooth,that is, it holds for all $x, x'$ that*

$$\|\nabla g(x) - \nabla g(x')\|^2 \leq \left(L + \frac{L^2}{\mu_y}\right)\|x - x'\|^2.$$

*Further, noting that $L/\mu_y \geq 1$, it implies that $g(x)$ is $(2L^2/\mu_y)$-smooth.*

**Lemma A.9** (Nouiehed et al. [32, Modified from Lemma A.3]). *Under Assumption 3.2 and 3.1, suppose $(x^*, y^*)$ is a saddle point of $f$. Denote the operator $y^*(\cdot)$ as the projection onto the optimal set of $\arg\max_{y \in \mathbb{R}^{d_y}} f(x, y)$, then it holds true that*

$$\|y^*(x) - y^*\|^2 \leq \frac{L^2}{\mu_y^2}\|x - x^*\|^2.$$

*Proof.* The proof is similar to the proof under strongly-convex-strongly-concave setting.

$$
\begin{aligned}
L^2\|x - x^*\|^2 &\geq \|\nabla_y f(x, y^*) - \nabla_y f(x^*, y^*)\|^2 \\
&= \|\nabla_y f(x, y^*)\|^2 \\
&\geq 2\mu_y\left(\max_y f(x, y) - f(x, y^*)\right) \\
&\geq \mu_y^2\|y^*(x) - y^*\|^2,
\end{aligned}
$$

where the first inequality is due to $L$-smooth of $f$, and the second line relies on $\nabla_y f(x^*, y^*) = 0$. The second inequality relies on PL condition in $y$ and in the last inequality we use the quadratic growth property by Lemma A.4. $\qquad\square$

# B  Two-timescale GDA matches AGDA

As a warm-up, we study GDA as well as AGDA with full gradient calculation in this section. After that, it is easy to extend the analysis when we are using a gradient estimator constructed by some variance reduction framework instead of the full gradient.

---

**Algorithm 3** AGDA $(f, (x_0, y_0), K, \tau_x, \tau_y)$

    **for** $k = 0, 1, \ldots, K - 1$ **do**

        $x_{k+1} = x_k - \tau_x \nabla_x f(x_k, y_k)$

        $y_{k+1} = y_k + \tau_y \nabla_y f(x_{k+1}, y_k)$

    **end for**

    **option I** (two-sided PL): **return** $(x_K, y_K)$

    **option II** (one-sided PL): **return** $(\hat{x}, \hat{y})$ chosen uniformly at random from $\{(x_k, y_k)\}_{k=0}^{K-1}$

---

**Algorithm 4** GDA $(f, (x_0, y_0), K, \tau_x, \tau_y)$

    **for** $k = 0, 1, \ldots, K - 1$ **do**

      $x_{k+1} = x_k - \tau_x \nabla_x f(x_k, y_k)$

      $y_{k+1} = y_k + \tau_y \nabla_y f(x_k, y_k)$

    **end for**

    **option I** (two-sided PL): **return** $(x_K, y_K)$

    **option II** (one-sided PL): **return** $(\hat{x}, \hat{y})$ chosen uniformly at random from $\{(x_k, y_k)\}_{k=0}^{K-1}$

## B.1 Convergence under Two-Sided PL condition

Under two-sided PL condition, it is known that AGDA [45] can find an $\epsilon$-approximate solution to a saddle point with a complexity of $\tilde{\mathcal{O}}(n\kappa_x \kappa_y^2 \log(1/\epsilon))$ when $\kappa_x \gtrsim \kappa_y$. However, the authors left us a question that whether GDA can converge under the same setting. We answer this question affirmatively in this section. We show that the same convergence rate can be achieved by GDA algorithm with simultaneous updates.

We define the following Lyapunov function as suggested by Doan [9]:

$$\mathcal{V}_k = \mathcal{A}_k + \frac{\lambda \tau_x}{\tau_y} \mathcal{B}_k,$$

where $\mathcal{A}_k = g(x_k) - g(x^*)$, $\mathcal{B}_k = g(x_k) - f(x_k, y_k)$. Then we can obtain the following statement.

**Theorem B.1.** *Suppose function $f(x, y)$ satisfies $L$-smooth, $\mu_x$-PL in $x$, $\mu_y$-PL in $y$. Let $\tau_y = 1/L$, $\lambda = 6L^2/\mu_y^2$ and $\tau_x = \tau_y/(22\lambda)$, then the sequence $\{(x_k, y_k)\}_{k=1}^K$ generated by Algorithm 4 satisfies:*

$$\mathcal{V}_{k+1} \le \left(1 - \frac{\mu_x \tau_x}{2}\right)^k \mathcal{V}_k.$$

*Proof.* Since we know that $g$ is $(2L^2/\mu_y)$- smooth by Lemma A.8, let $\tau_x \le \mu_y/(2L^2)$, we have

$$
\begin{aligned}
g(x_{k+1}) &\le g(x_k) - g(x^*) + \nabla g(x_k)^\top (x_{k+1} - x_k) + \frac{L^2}{\mu_y} \|x_{k+1} - x_k\|^2 \\
&\le g(x_k) - \tau_x \nabla g(x_k)^\top \nabla_x f(x_k, y_k) + \frac{\tau_x}{2} \|\nabla_x f(x_k, y_k)\|^2 \qquad (7) \\
&= g(x_k) - \frac{\tau_x}{2} \|\nabla g(x_k)\|^2 + \frac{\tau_x}{2} \|\nabla g(x_k) - \nabla_x f(x_k, y_k)\|^2,
\end{aligned}
$$

which implies

$$\mathcal{A}_{k+1} \le \mathcal{A}_k - \frac{\tau_x}{2} \|\nabla g(x_k)\|^2 + \frac{\tau_x}{2} \|\nabla g(x_k) - \nabla_x f(x_k, y_k)\|^2. \qquad (8)$$

Using the property of $L$-smooth, we know that the difference between $f(x_k, y_k)$ and $f(x_{k+1}, y_{k+1})$ can be bounded. Noting that $\tau_x \le 1/L$, we can obtain

$$
\begin{aligned}
f(x_k, y_k) - f(x_{k+1}, y_k) &\le -\nabla_x f(x_k, y_k)^\top (x_{k+1} - x_k) + \frac{L}{2} \|x_{k+1} - x_k\|^2 \\
&= \tau_x \|\nabla_x f(x_k, y_k)\|^2 + \frac{\tau_x^2 L}{2} \|\nabla_x f(x_k, y_k)\|^2 \qquad (9) \\
&\le \frac{3\tau_x}{2} \|\nabla_x f(x_k, y_k)\|^2.
\end{aligned}
$$

Let $\tau_y < 1/L$, then we have

$$f(x_{k+1}, y_k) - f(x_{k+1}, y_{k+1})$$

$$\leq - \nabla_y f(x_{k+1}, y_k)^\top (y_{k+1} - y_k) + \frac{L}{2}\|y_{k+1} - y_k\|^2$$

$$\leq - \tau_y \nabla_y f(x_{k+1}, y_k)^\top \nabla_y f(x_k, y_k) + \frac{\tau_y}{2}\|\nabla_y f(x_k, y_k)\|^2$$

$$= - \frac{\tau_y}{2}\|\nabla_y f(x_{k+1}, y_k)\|^2 + \frac{\tau_y}{2}\|\nabla_y f(x_k, y_k) - \nabla_y f(x_{k+1}, y_k)\|^2 \tag{10}$$

$$\leq - \frac{\tau_y}{4}\|\nabla_y f(x_k, y_k)\|^2 + \tau_y\|\nabla_y f(x_k, y_k) - \nabla_y f(x_{k+1}, y_k)\|^2$$

$$\leq - \frac{\tau_y}{4}\|\nabla_y f(x_k, y_k)\|^2 + \tau_y \tau_x^2 L^2 \|\nabla_x f(x_k, y_k)\|^2$$

$$\leq - \frac{\tau_y}{4}\|\nabla_y f(x_k, y_k)\|^2 + \tau_x\|\nabla_x f(x_k, y_k)\|^2,$$

where in the first inequality we use $f$ is $L$-smooth, and we use $\tau_y \leq 1/L$ in the second one and Young's inequality of $-\|a-b\|^2 \leq \frac{1}{2}\|a\|^2 + \|b\|^2$ in the third one.

Combing (9) and (11), we can see that

$$f(x_k, y_k) - f(x_{k+1}, y_{k+1}) \leq - \frac{\tau_y}{4}\|\nabla_y f(x_k, y_k)\|^2 + \frac{5\tau_x}{2}\|\nabla_x f(x_k, y_k)\|^2. \tag{11}$$

Now we can describe how $\mathcal{B}_{k+1}$ declines compared with $\mathcal{B}_k$, using (7) and (11), we have

$$\mathcal{B}_{k+1} = g(x_{k+1}) - g(x_k) + g(x_k) - f(x_k, y_k) + f(x_k, y_k) - f(x_{k+1}, y_{k+1})$$

$$\leq \mathcal{B}_k - \frac{\tau_x}{2}\|\nabla g(x_k)\|^2 + \frac{\tau_x}{2}\|\nabla g(x_k) - \nabla_x f(x_k, y_k)\|^2 \tag{12}$$

$$- \frac{\tau_y}{4}\|\nabla_y f(x_k, y_k)\|^2 + \frac{5\tau_x}{2}\|\nabla_x f(x_k, y_k)\|^2.$$

Using the inequality $\|\nabla_x f(x_k, y_k)\|^2 \leq 2\|g(x_k)\|^2 + 2\|\nabla g(x_k) - \nabla_x f(x_k, y_k)\|^2$, we have

$$\mathcal{B}_{k+1} \leq \mathcal{B}_k + \frac{9\tau_x}{2}\|\nabla g(x_k)\|^2 + \frac{11\tau_x}{2}\|\nabla g(x_k) - \nabla_x f(x_k, y_k)\|^2 - \frac{\tau_y}{4}\|\nabla_y f(x_k, y_k)\|^2.$$

By Lemma A.5, Lemma A.6 and Assumption 3.2, we have

$$\|\nabla g(x_k)\|^2 \geq 2\mu_x(g(x_k) - g(x^*)),$$

$$\|\nabla_x f(x_k, y_k) - \nabla g(x_k)\|^2 \leq \frac{2L^2}{\mu_y}(g(x_k) - f(x_k, y_k)), \tag{13}$$

$$\|\nabla_y f(x_k, y_k)\|^2 \geq 2\mu_y(g(x_k) - f(x_k, y_k)).$$

Since we let $\tau_y = 1/L$, $\lambda = 6L^2/\mu_y^2$ and $\tau_x = \tau_y/(22\lambda)$, we can obtain

$$\mathcal{V}_{k+1} = \mathcal{A}_{k+1} + \frac{\lambda \tau_x}{\tau_y}\mathcal{B}_{k+1}$$

$$\leq \mathcal{A}_k + \frac{\lambda \tau_x}{\tau_y}\mathcal{B}_k - \left(1 - \frac{9\lambda \tau_x}{\tau_y}\right)\frac{\tau_x}{2}\|\nabla g(x_k)\|^2$$

$$+ \left(1 + \frac{11\lambda \tau_x}{\tau_y}\right)\frac{\tau_x}{2}\|\nabla_x f(x_k, y_k) - \nabla g(x_k)\|^2 - \frac{\lambda \tau_x}{4}\|\nabla_y f(x_k, y_k)\|^2 \tag{14}$$

$$\leq \mathcal{A}_k - \left(1 - \frac{9\lambda \tau_x}{\tau_y}\right)\tau_x \mu_x \mathcal{A}_k + \frac{\lambda \tau_x}{\tau_y}\mathcal{B}_k + \left(1 + \frac{11\lambda \tau_x}{\tau_y}\right)\frac{\tau_x L^2}{\mu_y}\mathcal{B}_k - \frac{\lambda \tau_x \mu_y}{2}\mathcal{B}_k$$

$$\leq \left(1 - \frac{\mu_x \tau_x}{2}\right)\mathcal{A}_k + \left(1 - \frac{\mu_y \tau_y}{4}\right)\frac{\lambda \tau_x}{\tau_y}\mathcal{B}_k$$

$$\leq \left(1 - \frac{\mu_x \tau_x}{2}\right)\mathcal{V}_k,$$

where in the second inequality we use $11\lambda \tau_x/\tau_y \leq 1/2$ by the choices of $\tau_x, \tau_y$ and $\lambda$, while we use the fact that $3\tau_x L^2/\mu_y \leq \lambda \tau_x \mu_y/2$ in the third one and $\mu_x \tau_x \leq \mu_y \tau_y/2$ in the last one.

$\square$

Now we show that the convergence of $\mathcal{V}_k$ is sufficient to guarantee the convergence to a saddle point.

**Corollary B.1.** *Suppose function $f(x, y)$ satisfies $L$-smooth, $\mu_x$-PL in $x$, $\mu_y$-PL in $y$ and $\kappa_x \gtrsim \kappa_y$. Define $\tau_x, \tau_y$ as in Lemma B.1 ,then then the sequence $\{(x_k, y_k)\}_{k=1}^{K}$ generated by Algorithm 4 satisfies:*

$$\|x_k - x^*\|^2 + \|y_k - y^*\|^2 \leq \frac{2c^k}{(1 - \sqrt{c})^2} \max\left\{\frac{4}{\mu_x}, \frac{88}{\mu_y}\right\} \mathcal{V}_0. \tag{15}$$

*where $c = 1 - \mu_x \tau_x / 2$. Further, Algorithm 4 can find an $\epsilon$-saddle point with no more than $\mathcal{O}(n\kappa_x \kappa_y^2 \log(\kappa_x \kappa_y / \epsilon))$ stochastic first-order oracle calls.*

*Proof.* The proof is similar to the proof of Theorem 3.2 in [45].

By Lemma A.4 and the fact that $2\tau_x^2 L^2 \leq 1$, $\tau_x \leq \mu_y/(2L^2)$ and $\tau_y \leq 1/L$ by the choices of $\tau_x, \tau_y$, we can see that

$$
\begin{aligned}
&\|x_{k+1} - x_k\|^2 + \|y_{k+1} - y_k\|^2 \\
&= \tau_x^2 \|\nabla_x f(x_k, y_k)\|^2 + \tau_y^2 \|\nabla_y f(x_k, y_k)\|^2 \\
&= \tau_x^2 \|\nabla_x f(x_k, y_k)\|^2 + \tau_y^2 \|\nabla_y f(x_k, y_k) - \nabla_y f(x_k, y^*(x_k))\|^2 \\
&\leq \tau_x^2 \|\nabla_x f(x_k, y_k)\|^2 + \|y_k - y^*(x_k)\|^2 \\
&\leq 2\tau_x^2 \|\nabla g(x_k)\|^2 + 2\tau_x^2 \|\nabla g(x_k) - \nabla_x f(x_k, y_k)\|^2 + \|y_k - y^*(x_k)\|^2 \\
&\leq 2\|x_k - x^*\|^2 + 2\|y^*(x_k) - y_k\|^2 \\
&\leq \frac{4}{\mu_x} \mathcal{A}_k + \frac{4}{\mu_y} \mathcal{B}_k \\
&\leq \max\left\{\frac{4}{\mu_x}, \frac{88}{\mu_y}\right\} \mathcal{V}_k \\
&\leq \max\left\{\frac{4}{\mu_x}, \frac{88}{\mu_y}\right\} \left(1 - \frac{\mu_x \tau_x}{2}\right)^k \mathcal{V}_0,
\end{aligned}
\tag{16}
$$

where in the last inequality we use $\lambda \tau_x / \tau_y = 1/22$. Then we have

$$\|x_{k+1} - x_k\| + \|y_{k+1} - y_k\| \leq \left(1 - \frac{\mu_x \tau_x}{2}\right)^{k/2} \sqrt{2 \max\left\{\frac{4}{\mu_x}, \frac{88}{\mu_y}\right\} \mathcal{V}_0}.$$

For $n \geq k$, we obtain

$$
\begin{aligned}
\|x_n - x_k\| + \|y_n - y_k\| &\leq \sum_{i=k}^{n-1} \|x_{i+1} - x_i\|^2 + \|y_{i+1} - y_i\|^2 \\
&\leq \sqrt{2 \max\left\{\frac{4}{\mu_x}, \frac{88}{\mu_y}\right\} \mathcal{V}_0} \sum_{i=k}^{\infty} \left(1 - \frac{\mu_x \tau_x}{2}\right)^{i/2} \\
&\leq \frac{c^{k/2}}{1 - \sqrt{c}} \sqrt{2 \max\left\{\frac{4}{\mu_x}, \frac{88}{\mu_y}\right\} \mathcal{V}_0},
\end{aligned}
$$

where $c = 1 - \mu_x \tau_x / 2$. We know that when $n \to \infty$, we have $(x_n, y_n) \to (x^*, y^*)$ where $(x^*, y^*)$ is a saddle point, Taking square on both sides completes our proof.

$\square$

## B.2 Convergence under One-sided PL condition

When $f$ is nonconvex in $x$, we have the following theorem for GDA.

**Theorem B.2.** *Suppose function $f(x, y)$ satisfies $L$-smooth, $\mu_y$-PL in $y$. Let $\tau_y = 1/L$, $\lambda = 4L^2/\mu_y^2$ and $\tau_x = \tau_y/(18\lambda)$, then the sequence $\{(x_k, y_k)\}_{k=0}^{K-1}$ generated by Algorithm 4 satisfies,*

$$\frac{1}{K} \sum_{k=0}^{K-1} \|\nabla g(x_k)\|^2 \leq \frac{288 L^3}{K \mu_y^2} \mathcal{V}_0.$$

*Furthermore, if we choose the output $(\hat{x}, \hat{y})$ uniformly from $\{(x_k, y_k)\}_{k=0}^{K-1}$, then we can get $\|\nabla g(\hat{x})\| \leq \epsilon$ with no more than $\mathcal{O}(n\kappa_y^2 L/\epsilon^2)$ first-order oracle calls.*

*Proof.* Using equation (8) and Lemma A.6 that $\|\nabla g(x_k) - \nabla_x f(x_k, y_k)\|^2 \leq 2L^2 \mathcal{B}_k/\mu_y$, we have

$$\mathcal{A}_{k+1} \leq \mathcal{A}_k - \frac{\tau_x}{2}\|\nabla g(x_k)\|^2 + \frac{\tau_x L^2}{\mu_y}\mathcal{B}_k. \tag{17}$$

Further, using equation (12), we have

$$\begin{aligned}
\mathcal{B}_{k+1} &\leq \mathcal{B}_k + \frac{9\tau_x}{2}\|\nabla g(x_k)\|^2 + \frac{11\tau_x}{2}\|\nabla g(x_k) - \nabla_x f(x_k, y_k)\|^2 - \frac{\tau_y}{4}\|\nabla_y f(x_k, y_k)\|^2 \\
&\leq \mathcal{B}_k + \frac{9\tau_x}{2}\|\nabla g(x_k)\|^2 + \frac{11\tau_x L^2}{\mu_y}\mathcal{B}_k - \frac{\mu_y \tau_y}{2}\mathcal{B}_k \\
&\leq (1 - \frac{\mu_y \tau_y}{4})\mathcal{B}_k + \frac{9\tau_x}{2}\|\nabla g(x_k)\|^2,
\end{aligned} \tag{18}$$

where we use Lemma A.6 and PL condition in $y$ in the first inequality and $11\tau_x L^2/\mu_y \leq \mu_y \tau_y/4$ by the choices of $\tau_x, \tau_y$. Thus,

$$\mathcal{B}_k \leq \left(1 - \frac{\mu_y \tau_y}{4}\right)^k \mathcal{B}_0 + \frac{9\tau_x}{2}\sum_{i=0}^{k-1}\left(1 - \frac{\mu_y \tau_y}{4}\right)^{k-i-1}\|\nabla g(x_i)\|^2.$$

Plugging into (17),

$$\mathcal{A}_{k+1} \leq \mathcal{A}_k - \frac{\tau_x}{2}\|\nabla g(x_k)\|^2 + \frac{\tau_x L^2}{\mu_y}\left(1 - \frac{\mu_y \tau_y}{4}\right)^k \mathcal{B}_0 + \frac{9\tau_x^2 L^2}{2\mu_y}\sum_{i=0}^{k-1}\left(1 - \frac{\mu_y \tau_y}{4}\right)^{k-i-1}\|\nabla g(x_i)\|^2.$$

Telescoping and noticing that $18\tau_x^2 L^2/\tau_y \mu_y^2 \leq \tau_x/4$ and $\lambda = 4L^2/\mu_y^2$, we have

$$\begin{aligned}
\mathcal{A}_{K+1} &\leq \mathcal{A}_0 - \frac{\tau_x}{2}\sum_{k=0}^{K}\|\nabla g(x_k)\|^2 + \frac{\tau_x L^2}{\mu_y}\sum_{k=0}^{K-1}\left(1 - \frac{\mu_y \tau_y}{4}\right)^k \mathcal{B}_0 \\
&\quad + \frac{9\tau_x^2 L^2}{2\mu_y}\sum_{k=1}^{K}\sum_{i=0}^{k-1}\left(1 - \frac{\mu_y \tau_y}{4}\right)^{k-i-1}\|\nabla g(x_i)\|^2 \\
&= \mathcal{A}_0 - \frac{\tau_x}{2}\sum_{k=0}^{K}\|\nabla g(x_k)\|^2 + \frac{\tau_x L^2}{\mu_y}\sum_{k=0}^{K-1}\left(1 - \frac{\mu_y \tau_y}{4}\right)^k \mathcal{B}_0 \\
&\quad + \frac{9\tau_x^2 L^2}{2\mu_y}\sum_{i=0}^{K-1}\sum_{k=i+1}^{K}\left(1 - \frac{\mu_y \tau_y}{4}\right)^{k-i-1}\|\nabla g(x_i)\|^2 \\
&\leq \mathcal{A}_0 - \frac{\tau_x}{2}\sum_{k=0}^{K}\|\nabla g(x_k)\|^2 + \frac{\tau_x L^2}{\mu_y}\sum_{k=0}^{K-1}\left(1 - \frac{\mu_y \tau_y}{4}\right)^k \mathcal{B}_0 + \frac{18\tau_x^2 L^2}{\tau_y \mu_y^2}\sum_{i=0}^{K-1}\|\nabla g(x_i)\|^2 \\
&\leq \mathcal{A}_0 - \frac{\tau_x}{2}\sum_{k=0}^{K}\|\nabla g(x_k)\|^2 + \frac{\tau_x L^2}{\mu_y}\sum_{k=0}^{K-1}\left(1 - \frac{\mu_y \tau_y}{4}\right)^k \mathcal{B}_0 + \frac{18\tau_x^2 L^2}{\tau_y \mu_y^2}\sum_{k=0}^{K}\|\nabla g(x_k)\|^2 \\
&\leq \mathcal{A}_0 - \frac{\tau_x}{4}\sum_{k=0}^{K}\|\nabla g(x_k)\|^2 + \frac{4\tau_x L^2}{\tau_y \mu_y^2}\mathcal{B}_0 \\
&= \mathcal{V}_0 - \frac{\tau_x}{4}\sum_{k=0}^{K}\|\nabla g(x_k)\|^2.
\end{aligned}$$

Rearranging and noticing that $\mathcal{A}_{K+1} \geq 0$, we can see that

$$\frac{1}{K+1}\sum_{k=0}^{K}\|\nabla g(x_k)\|^2 \leq \frac{4\mathcal{V}_0}{(K+1)\tau_x},$$

which is equivalent to the desired inequality. $\qquad\square$

**Algorithm 5** SVRG-GDA $(f, (x_0, y_0), T, S, M, B, \tau_x, \tau_y)$

1: $\bar{x}_0 = x_0, \bar{y}_0 = y_0$
2: **for** $t = 0, 1, \ldots, T - 1$ **do**
3:     **for** $s = 0, 1, \ldots, S - 1$ **do**
4:         $x_{s,0} = \bar{x}_s, y_{s,0} = \bar{y}_s$
5:         compute $\nabla_x f(\bar{x}_s, \bar{y}_s) = \frac{1}{n} \sum_{i=1}^n \nabla_x f_i(\bar{x}_s, \bar{y}_s)$
6:         compute $\nabla_y f(\bar{x}_s, \bar{y}_s) = \frac{1}{n} \sum_{i=1}^n \nabla_y f_i(\bar{x}_s, \bar{y}_s)$
7:         **for** $k = 0, 1, \ldots, M - 1$
8:             draw samples $S_x, S_y$ independently with both size $B$.
9:             $G_x(x_{s,k}, y_{s,k}) = \frac{1}{B} \sum_{i \in S_x} [\nabla_x f_i(x_{s,k}, y_{s,k}) - \nabla_x f_i(\bar{x}_s, \bar{y}_s) + \nabla_x f(\bar{x}_s, \bar{y}_s)]$
10:            $G_y(x_{s,k}, y_{s,k}) = \frac{1}{B} \sum_{i \in S_y} [\nabla_y f_i(x_{s,k}, y_{s,k}) - \nabla_x f_i(\bar{x}_s, \bar{y}_s) + \nabla_x f(\bar{x}_s, \bar{y}_s)]$
11:            $x_{s,k+1} = x_{s,k} - \tau_x G_x(x_{s,k}, y_{s,k})$
12:            $y_{s,k+1} = y_{s,k} + \tau_y G_y(x_{s,k}, y_{s,k})$
13:         **end for**
14:         $\bar{x}_{s+1} = x_{s,M}, \bar{y}_{s+1} = y_{s,M}$
15:     **end for**
16:     choose $(x_t, y_t)$ from $\{\{(x_{s,k}, y_{s,k})\}_{k=0}^{M-1}\}_{s=0}^{S-1}$ uniformly at random.
17:     $\bar{x}_0 = x_t, \bar{y}_0 = y_t$
18: **end for**
19: **return** $(x_T, y_T)$

## C   Convergence of GDA with SVRG Gradient Estimators

In this section, we will show the convergence rate of GDA with SVRG gradient estimators is $\mathcal{O}((n + n^{2/3} \kappa_x \kappa_y^2) \log(1/\epsilon))$, which is faster than the result of $\mathcal{O}((n + n^{2/3} \max\{\kappa_x^3, \kappa_y^3\}) \log(1/\epsilon))$ given by SVRG-AGDA Yang et al. [45]. Plus, we can prove that the same convergence rate can be achieved by AGDA with similar techniques since in our algorithm we set $\tau_x \ll \tau_y$, therefore $x_k$ changes much slower than $y_k$. The reason for unbalanced step sizes lies in that $g(x)$ is $(L + L^2/\mu_y)$-smooth by Lemma A.8. Thus, we can regard that the condition number of solving problem $\max_{y \in \mathbb{R}^{d_y}} f(x, y)$ is $\mathcal{O}(\kappa_y)$ while that of solving $\min_{x \in \mathbb{R}^{d_x}} g(x)$ is $\mathcal{O}(\kappa_x \kappa_y)$. Thus, it is reasonable that the total complexity has a factor of $\mathcal{O}(\kappa_x \kappa_y^2)$. The algorithm is described in 5.

For the innermost loop about subscript $k$ when $t$ and $s$ are both fixed, we define the Lyapunov function as follows:

$$\mathcal{V}_{s,k} = \mathcal{A}_{s,k} + \frac{\lambda \tau_x}{\tau_y} \mathcal{B}_{s,k} + c_{s,k} \|x_{s,k} - \bar{x}_s\|^2 + d_{s,k} \|y_k - \bar{y}_s\|^2,$$

where $\mathcal{A}_{s,k} = g(x_{s,k}) - g(x^*)$ and $\mathcal{B}_{s,k} = g(x_{s,k}) - f(x_{s,k}, y_{s,k})$ and $c_{s,k}, d_{s,k}$ will be defined recursively with $c_{s,M} = d_{s,M} = 0$ in our proof. Then we can have the following lemma.

**Lemma C.1.** *Under Assumption 6.1 and 3.1, if we let $\tau_y = \nu/(Ln^\alpha)$, $\lambda = 14L^2/\mu_y^2$ and $\tau_x = \tau_y/(22\lambda)$, where $\nu = 1/(176(\mathrm{e} - 1)), 0 < \alpha \leq 1$; let $B = 1, M = \lfloor n^{3\alpha/2}/(2\nu) \rfloor$. Then for Algorithm 5, the following statement holds true:*

$$\mathbb{E}[\mathcal{V}_{s,k+1}] \leq \mathcal{V}_{s,k} - \frac{\tau_x}{8} \|\nabla g(x_{s,k})\|^2 - \frac{\lambda \tau_x}{16} \|\nabla_y f(x_{s,k}, y_{s,k})\|^2,$$

*where $\mathcal{A}_{s,k} = g(x_{s,k}) - g(x^*)$ and $\mathcal{B}_{s,k} = g(x_{s,k}) - f(x_{s,k}, y_{s,k})$. Above, the definitions of $c_{s,k}, d_{s,k}$ is given recursively with $c_{s,M} = d_{s,M} = 0$ as:*

$$c_{s,k} = c_{s,k+1}(1 + \tau_x \gamma_1) + \left(c_{s,k+1} \tau_x^2 + \frac{3\tau_x^2 L^2}{\mu_y}\right) L^2 + \left(d_{s,k+1} \tau_y^2 + \frac{\lambda \tau_x \tau_y L}{2}\right) L^2,$$

$$d_{s,k} = d_{s,k+1}(1 + \tau_y \gamma_2) + \left(c_{s,k+1}\tau_x^2 + \frac{3\tau_x^2 L^2}{\mu_y}\right)L^2 + \left(d_{s,k+1}\tau_y^2 + \frac{\lambda \tau_x \tau_y L}{2}\right)L^2.$$

*Proof.* Since $s$ is fixed in thie lemma, we omit subscripts of $t$ in the following proofs, then the Lyapunov function can be written as:

$$\mathcal{V}_k = \mathcal{A}_k + \frac{\lambda \tau_x}{\tau_y}\mathcal{B}_k + c_k \|x_k - \bar{x}\|^2 + d_k \|y_k - \bar{y}\|^2.$$

Before the formal proof, we present some standard properties of variance reduction. We denote the stochastic gradients as:

$$G_x(x_k, y_k) = \frac{1}{B}\sum_{i \in S_x}\left(\nabla_x f_i(x_k, y_k) - \nabla_x f_i(\bar{x}, \bar{y}) + \nabla_x f(\bar{x}, \bar{y})\right)$$

and

$$G_y(x_k, y_k) = \frac{1}{B}\sum_{i \in S_y}\left(\nabla_y f_i(x_k, y_k) - \nabla_y f_i(\bar{x}, \bar{y}) + \nabla_y f(\bar{x}, \bar{y})\right).$$

Then we know that the stochastic gradients satisfy unbiasedness that

$$\mathbb{E}[G_x(x_k, y_k)] = \nabla_x f(x_k, y_k) \quad \text{and} \quad \mathbb{E}[G_y(x_k, y_k)] = \nabla_y f(x_k, y_k).$$

And we can bound the variance of the stochastic gradients as follows:

$$\begin{aligned}
&\mathbb{E}\|G_x(x_k, y_k) - \nabla_x f(x_k, y_k)\|^2 \\
&= \mathbb{E}\|\nabla_x f_i(x_k, y_k) - \nabla_x f_i(\bar{x}, \bar{y}) + \nabla_x f(\bar{x}, \bar{y}) - \nabla_x f(x_k, y_k)\|^2 \\
&\leq \mathbb{E}\|\nabla_x f_i(x_k, y_k) - \nabla_x f_i(\bar{x}, \bar{y})\|^2 \\
&\leq L^2 \mathbb{E}\|x_k - \bar{x}\|^2] + L^2 \mathbb{E}[\|y_k - \bar{y}\|^2.
\end{aligned} \tag{19}$$

Similarly, we have

$$\mathbb{E}\|G_y(x_k, y_k) - \nabla_y f(x_k, y_k)\|^2 \leq L^2 \mathbb{E}\|x_k - \bar{x}\|^2 + L^2 \mathbb{E}\|y_k - \bar{y}\|^2. \tag{20}$$

Equipped with the above properties of SVRG, now we can begin our proof of Lemma C.1.

Since we know that $g$ is $(2L^2/\mu_y)$-smooth by Lemma A.8 and $\tau_x \leq \mu_y/2L^2$, we have

$$\begin{aligned}
\mathbb{E}[g(x_{k+1})] &\leq \mathbb{E}\left[g(x_k) + \nabla g(x_k)^\top(x_{k+1} - x_k) + \frac{2L^2}{\mu_y}\|x_{k+1} - x_k\|^2\right] \\
&\leq \mathbb{E}\left[g(x_k) - \tau_x \nabla g(x_k)^\top G_x(x_k, y_k) + \frac{\tau_x^2 L^2}{\mu_y}\|G_x(x_k, y_k)\|^2\right] \\
&\leq \mathbb{E}\left[g(x_k) - \tau_x \nabla g(x_k)^\top \nabla_x f(x_k, y_k) + \frac{\tau_x}{2}\|\nabla_x f(x_k, y_k)\|^2\right] \\
&\quad + \mathbb{E}\left[\frac{\tau_x^2 L^2}{\mu_y}\|G_x(x_k, y_k) - \nabla_x f(x_k, y_k)\|^2\right] \\
&= \mathbb{E}\left[g(x_k) - \frac{\tau_x}{2}\|\nabla g(x_k)\|^2 + \frac{\tau_x}{2}\|\nabla g(x_k) - \nabla_x f(x_k, y_k)\|^2\right] \\
&\quad + \mathbb{E}\left[\frac{\tau_x^2 L^2}{\mu_y}\|G_x(x_k, y_k) - \nabla_x f(x_k, y_k)\|^2\right],
\end{aligned} \tag{21}$$

where we use $\tau_x^2 L^2 \leq \mu_y$ in the third inequality. Also, we have

$$\begin{aligned}
\mathbb{E}[f(x_k, y_k)] &\leq \mathbb{E}\left[f(x_{k+1}, y_k) - \nabla_x f(x_k, y_k)^\top(x_{k+1} - x_k) + \frac{L}{2}\|x_{k+1} - x_k\|^2\right] \\
&= \mathbb{E}\left[f(x_{k+1}, y_k) + \tau_x \nabla_x f(x_k, y_k)^\top G_x(x_k, y_k) + \frac{\tau_x^2 L}{2}\|G_x(x_k, y_k)\|^2\right] \\
&= \mathbb{E}\left[f(x_{k+1}, y_k) + \tau_x \|\nabla_x f(x_k, y_k)\|^2 + \frac{\tau_x^2 L}{2}\|\nabla_x f(x_k, y_k)\|^2\right]
\end{aligned}$$

$$+ \mathbb{E}\left[\frac{\tau_x^2 L}{2}\|G_x(x_k, y_k) - \nabla_x f(x_k, y_k)\|^2\right]$$

$$\leq \mathbb{E}\left[f(x_{k+1}, y_k) + \frac{3\tau_x}{2}\|\nabla_x f(x_k, y_k)\|^2 + \frac{\tau_x^2 L}{2}\|G_x(x_k, y_k) - \nabla_x f(x_k, y_k)\|^2\right],$$

where we use the quadratic upper bound implied by $L$-smoothness in the first inequality and $\tau_y \leq 1/L$ in the second one. Similarly,

$$\mathbb{E}[f(x_{k+1}, y_k)] \leq \mathbb{E}\left[f(x_{k+1}, y_{k+1}) - \nabla_y f(x_{k+1}, y_k)^\top (y_{k+1} - y_k) + \frac{L}{2}\|y_{k+1} - y_k\|^2\right]$$

$$= \mathbb{E}\left[f(x_{k+1}, y_{k+1}) - \tau_y \nabla_y f(x_{k+1}, y_k)^\top G_y(x_k, y_k) + \frac{\tau_y^2 L}{2}\|G_y(x_k, y_k)\|^2\right]$$

$$\leq \mathbb{E}\left[f(x_{k+1}, y_{k+1}) - \tau_y \nabla_y f(x_{k+1}, y_k)^\top \nabla_y f(x_k, y_k) + \frac{\tau_y}{2}\|\nabla_y f(x_k, y_k)\|^2\right]$$

$$+ \mathbb{E}\left[\frac{\tau_y^2 L}{2}\|G_y(x_k, y_k) - \nabla_y f(x_k, y_k)\|^2\right]$$

$$= \mathbb{E}\left[f(x_{k+1}, y_{k+1}) - \frac{\tau_y}{2}\|\nabla_y f(x_{k+1}, y_k)\|^2\right]$$

$$+ \mathbb{E}\left[\frac{\tau_y}{2}\|\nabla_y f(x_{k+1}, y_k) - \nabla_y f(x_k, y_k)\|^2 + \frac{\tau_y^2 L}{2}\|G_y(x_k, y_k) - \nabla_y f(x_k, y_k)\|^2\right]$$

$$\leq \mathbb{E}\left[f(x_{k+1}, y_{k+1}) - \frac{\tau_y}{4}\|\nabla_y f(x_{k+1}, y_k)\|^2\right]$$

$$+ \mathbb{E}\left[\tau_y\|\nabla_y f(x_{k+1}, y_k) - \nabla_y f(x_k, y_k)\|^2 + \frac{\tau_y^2 L}{2}\|G_y(x_k, y_k) - \nabla_y f(x_k, y_k)\|^2\right]$$

$$\leq \mathbb{E}\left[f(x_{k+1}, y_{k+1}) - \frac{\tau_y}{4}\|\nabla_y f(x_{k+1}, y_k)\|^2\right]$$

$$+ \mathbb{E}\left[\tau_y \tau_x^2 L^2 \|G_x(x_k, y_k)\|^2 + \frac{\tau_y^2 L}{2}\|G_y(x_k, y_k) - \nabla_y f(x_k, y_k)\|^2\right]$$

$$\leq \mathbb{E}\left[f(x_{k+1}, y_{k+1}) - \frac{\tau_y}{4}\|\nabla_y f(x_{k+1}, y_k)\|^2 + \tau_x\|\nabla_x f(x_k, y_k)\|^2\right]$$

$$+ \mathbb{E}\left[\tau_x^2 L \|G_x(x_k, y_k) - \nabla_x f(x_k, y_k)\|^2 + \frac{\tau_y^2 L}{2}\|G_y(x_k, y_k) - \nabla_y f(x_k, y_k)\|^2\right].$$

Above, the first and fourth inequalities are both due to $L$-smoothness; the second one follows from $\tau_y \leq 1/L$; the third one uses the fact that $-\mathbb{E}[\|a - b\|^2] \leq -\frac{1}{2}\mathbb{E}[\|a\|^2] + \mathbb{E}[\|b\|^2]$; the last one relies on $\mathbb{E}[\|G_x(x_k, y_k)\|^2] = \mathbb{E}[\|\nabla_x f(x_k, y_k)\|^2 + \|G_x(x_k, y_k) - \nabla_x f(x_k, y_k)\|^2]$ and the choices of $\tau_x, \tau_y$. Summing up the above two inequalities, we obtain

$$\mathbb{E}[f(x_k, y_k)] \leq \mathbb{E}\left[f(x_{k+1}, y_{k+1}) + \frac{5\tau_x}{2}\|\nabla_x f(x_k, y_k)\|^2 + \frac{3\tau_x^2 L}{2}\|G_x(x_k, y_k) - \nabla_x f(x_k, y_k)\|^2\right]$$

$$\mathbb{E}\left[-\frac{\tau_y}{4}\|\nabla_y f(x_k, y_k)\|^2 + \frac{\tau_y^2 L}{2}\|G_y(x_k, y_k) - \nabla_y f(x_k, y_k)\|^2\right].$$

Combing with inequality (21), we can see that

$$\mathbb{E}[\mathcal{B}_{k+1}] \leq \mathbb{E}\left[\mathcal{B}_k - \frac{\tau_x}{2}\|\nabla g(x_k)\|^2 + \frac{\tau_x}{2}\|\nabla g(x_k) - \nabla_x f(x_k, y_k)\|^2\right]$$

$$+ \mathbb{E}\left[-\frac{\tau_y}{4}\|\nabla_y f(x_k, y_k)\|^2 + \frac{5\tau_x}{2}\|\nabla_x f(x_k, y_k)\|^2\right]$$

$$+ \mathbb{E}\left[\frac{\tau_y^2 L}{2}\|G_y(x_k, y_k) - \nabla_y f(x_k, y_k)\|^2 + \frac{5\tau_x^2 L^2}{2\mu_y}\|G_x(x_k, y_k) - \nabla_x f(x_k, y_k)\|^2\right]$$

$$\leq \mathbb{E}\left[\mathcal{B}_k + \frac{9\tau_x}{2}\|\nabla g(x_k)\|^2 + \frac{11\tau_x}{2}\|\nabla g(x_k) - \nabla_x f(x_k, y_k)\|^2\right]$$

$$+ \mathbb{E}\left[-\frac{\tau_y}{4}\|\nabla_y f(x_k, y_k)\|^2\right]$$

$$+ \mathbb{E}\left[\frac{\tau_y^2 L}{2}\|G_y(x_k, y_k) - \nabla_y f(x_k, y_k)\|^2 + \frac{5\tau_x^2 L^2}{2\mu_y}\|G_x(x_k, y_k) - \nabla_x f(x_k, y_k)\|^2\right],$$

$$(22)$$

where we use $\mathbb{E}\|\nabla_x f(x_k, y_k)\|^2 \leq \mathbb{E}\|\nabla g(x_k)\|^2 + \mathbb{E}\|\nabla g(x_k) - \nabla_x f(x_k, y_k)\|^2$. Using Young's inequality as equation (37) and (38) in [45], we have

$$\mathbb{E}\|x_{k+1} - \bar{x}\|^2 \leq \mathbb{E}\left[\tau_x^2 \|G_x(x_k, y_k) - \nabla_x f(x_k, y_k)\|^2\right]$$
$$+ \mathbb{E}\left[(1 + \tau_x \gamma_1)\|x_k - \bar{x}\|^2 + \left(\tau_x^2 + \frac{\tau_x}{\gamma_1}\right)\|\nabla_x f(x_k, y_k)\|^2\right],$$

$$\mathbb{E}\|y_{k+1} - \bar{y}\|^2 \leq \mathbb{E}\left[\tau_y^2 \|G_y(x_k, y_k) - \nabla_y f(x_k, y_k)\|^2\right]$$
$$+ \mathbb{E}\left[(1 + \tau_y \gamma_2)\|y_k - \bar{y}\|^2 + \left(\tau_y^2 + \frac{\tau_y}{\gamma_2}\right)\|\nabla_y f(x_k, y_k)\|^2\right],$$

where $\gamma_1, \gamma_2$ are two positive constant in Young's inequality which will be chosen later.

Then, using equation (19), (20), (21), (22), we have

$$\mathbb{E}[\mathcal{V}_{k+1}] = \mathbb{E}\left[\mathcal{A}_{k+1} + \frac{\lambda \tau_x}{\tau_y}\mathcal{B}_{k+1} + c_{k+1}\|x_{k+1} - \bar{x}\|^2 + d_{k+1}\|y_{k+1} - \bar{y}\|^2\right]$$

$$\leq \mathcal{A}_k + \frac{\lambda \tau_x}{\tau_y}\mathcal{B}_k - \left(1 - \frac{9\lambda \tau_x}{\tau_y}\right)\frac{\tau_x}{2}\|\nabla g(x_k)\|^2 +$$

$$+ \left(1 + \frac{11\lambda \tau_x}{\tau_y}\right)\frac{\tau_x}{2}\|\nabla_x f(x_k, y_k) - \nabla g(x_k)\|^2$$

$$- \frac{\lambda \tau_x}{4}\|\nabla_y f(x_k, y_k)\|^2 + \frac{2\tau_x^2 L^2}{\mu_y}\left(1 + \frac{5\lambda \tau_x}{4\tau_y}\right)\|G_x(x_k, y_k) - \nabla_x f(x_k, y_k)\|^2$$

$$+ \frac{\lambda \tau_x \tau_y L}{2}\|G_y(x_k, y_k) - \nabla_y f(x_k, y_k)\|^2$$

$$+ \mathbb{E}[c_{k+1}\|x_{k+1} - \bar{x}\|^2 + d_{k+1}\|y_{k+1} - \bar{y}\|^2]$$

$$= \mathcal{V}_k - \left(1 - \frac{9\lambda \tau_x}{\tau_y}\right)\frac{\tau_x}{2}\|\nabla g(x_k)\|^2 + \left(1 + \frac{11\lambda \tau_x}{\tau_y}\right)\frac{\tau_x}{2}\|\nabla_x f(x_k, y_k) - \nabla g(x_k)\|^2$$

$$- \frac{\lambda \tau_x}{4}\|\nabla_y f(x_k, y_k)\|^2 + c_{k+1}\left(\tau_x^2 + \frac{\tau_x}{\gamma_1}\right)\|\nabla_x f(x_k, y_k)\|^2$$

$$+ d_{k+1}\left(\tau_y^2 + \frac{\tau_y}{\gamma_2}\right)\|\nabla_y f(x_k, y_k)\|^2$$

$$\leq \mathcal{V}_k - \left(1 - \frac{9\lambda \tau_x}{\tau_y}\right)\frac{\tau_x}{2}\|\nabla g(x_k)\|^2 + \left(1 + \frac{11\lambda \tau_x}{\tau_y}\right)\frac{\tau_x}{2}\|\nabla_x f(x_k, y_k) - \nabla g(x_k)\|^2$$

$$- \frac{\lambda \tau_x}{4}\|\nabla_y f(x_k, y_k)\|^2 + d_{k+1}\left(\tau_y^2 + \frac{\tau_y}{\gamma_2}\right)\|\nabla_y f(x_k, y_k)\|^2$$

$$+ 2c_{k+1}\left(\tau_x^2 + \frac{\tau_x}{\gamma_1}\right)\|\nabla g(x_k)\|^2$$

$$+ 2c_{k+1}\left(\tau_x^2 + \frac{\tau_x}{\gamma_1}\right)\|\nabla_x f(x_k, y_k) - \nabla g(x_k)\|^2$$

$$\leq \mathcal{V}_k - \frac{\tau_x}{4}\|\nabla g(x_k)\|^2 + \frac{3\tau_x}{4}\|\nabla_x f(x_k, y_k) - \nabla g(x_k)\|^2 - \frac{\lambda \tau_x}{4}\|\nabla_y f(x_k, y_k)\|^2$$

$$+ d_{k+1}\left(\tau_y^2 + \frac{\tau_y}{\gamma_2}\right)\|\nabla_y f(x_k, y_k)\|^2 + 2c_{k+1}\left(\tau_x^2 + \frac{\tau_x}{\gamma_1}\right)\|\nabla g(x_k)\|^2$$

$$+ 2c_{k+1}\left(\tau_x^2 + \frac{\tau_x}{\gamma_1}\right)\|\nabla_x f(x_k, y_k) - \nabla g(x_k)\|^2,$$

$$(23)$$

where the second last inequality relies on

$$\|\nabla_x f(x_k, y_k)\|^2 \leq 2\|\nabla g(x_k)\|^2 + 2\|\nabla g(x_k) - \nabla_x f(x_k, y_k)\|^2;$$

in the last inequality we use $11\lambda \tau_x / \tau_y \leq 1/2$ by our choices of $\lambda, \tau_x, \tau_y$; the second equality is due to the definition of $c_{k+1}, d_{k+1}$.

Now we define $e_k = \max\{c_k, d_k\}$ and we bound $e_k$ by letting $\gamma_1 = \lambda L/n^{\alpha/2}$ and $\gamma_2 = L/n^{\alpha/2}$. Then according to the definition of $c_k, d_k$ given by our definition, we have

$$e_k \leq (1 + \tau_y\gamma_2 + \tau_y^2 L^2)e_{k+1} + \frac{3\tau_x^2 L^4}{\mu_y} + \frac{\lambda\tau_x\tau_y L^3}{2}$$

$$\leq (1 + \tau_y\gamma_2 + \tau_y^2 L^2)e_{k+1} + \tau_y^2 L^3$$

$$= \left(1 + \frac{\nu}{n^{3\alpha/2}} + \frac{\nu^2}{n^{2\alpha}}\right) + \frac{L\nu^2}{n^{2\alpha}}$$

$$\leq \left(1 + \frac{2\nu}{n^{3\alpha/2}}\right)e_{k+1} + \frac{L\nu^2}{n^{2\alpha}},$$

where we use $\tau_x \leq \tau_y$ and $\gamma_1\tau_x \leq \gamma_2\tau_y$ in the first; the second inequality is due to $\tau_x^2 L/\mu_y \leq \tau_y^2/6$ and $\lambda\tau_x \leq \tau_y/4$ ; we plug in $\tau_y = \nu/Ln^\alpha$ in the third line; we use $\nu \leq 1$ in the last inequality.

Since $M = \lfloor n^{3\alpha/2}/(2\nu) \rfloor$ and $c_M = d_M = 0$, if we define $\theta = 2\nu/n^{3\alpha/2}$, then

$$e_0 \leq \frac{L\nu^2}{n^{2\alpha}} \frac{(1+\theta)^M - 1}{\theta} \leq \frac{L\nu(e-1)}{2n^{\alpha/2}}. \tag{24}$$

Since $e_{k+1} \leq e_k$, we know that $e_k \leq e_0$, then

$$d_{k+1}\left(\tau_y^2 + \frac{\tau_y}{\gamma_2}\right) \leq e_0\left(\tau_y + \frac{1}{\gamma_2}\right)\tau_y$$

$$\leq \frac{L\nu(e-1)}{2n^{\alpha/2}}\left(\tau_y + \frac{1}{\gamma_2}\right)\tau_y \tag{25}$$

$$= \frac{\nu(e-1)}{2}\left(\frac{\nu}{n^{3\alpha/2}} + 1\right)\tau_y$$

$$\leq \nu(e-1)\tau_y,$$

where we use $d_{k+1} \leq e_0$ in the first inequality and (24) in the second one, and in the third line we plug in $\tau_y = \nu/(Ln^\alpha)$ and $\gamma_2 = L/n^{\alpha/2}$. The last inequality follows from $\nu/n^{3\alpha/2} \leq 1$.

Similarly, note that $\gamma_1 \geq \gamma_2$ and $\tau_x \leq \tau_y$ by the choices of $\tau_x, \tau_y$, then we have

$$c_{k+1}\left(\tau_x^2 + \frac{\tau_x}{\gamma_1}\right) \leq e_0\left(\tau_x + \frac{1}{\gamma_1}\right)\tau_x \leq e_0\left(\tau_y + \frac{1}{\gamma_2}\right)\tau_x \leq 3\nu(e-1)\tau_x. \tag{26}$$

Plugging (25) and (26) into (23),

$$\mathbb{E}[\mathcal{V}_{k+1}] \leq \mathcal{V}_k - \frac{\tau_x}{4}\|\nabla g(x_k)\|^2 + \frac{3\tau_x}{4}\|\nabla_x f(x_k, y_k) - \nabla g(x_k)\|^2 - \frac{\lambda\tau_x}{4}\|\nabla_y f(x_k, y_k)\|^2$$

$$+ \nu(e-1)\tau_y\|\nabla_y f(x_k, y_k)\|^2 + 2\nu(e-1)\tau_x\|\nabla g(x_k)\|^2$$

$$+ 2\nu(e-1)\tau_x\|\nabla_x f(x_k, y_k) - \nabla g(x_k)\|^2.$$

If we let $\nu \leq 1/(176(e-1))$, then we can verify that the following statements hold true:

$$\nu(e-1)\tau_y \leq \frac{\lambda\tau_x}{8},$$

$$2\nu(e-1)\tau_x \leq \frac{\tau_x}{8}.$$

Thus,

$$\mathbb{E}[\mathcal{V}_{k+1}] \leq \mathcal{V}_k - \frac{\tau_x}{8}\|\nabla g(x_k)\|^2 + \frac{7\tau_x}{8}\|\nabla_x f(x_k, y_k) - \nabla g(x_k)\|^2 - \frac{\lambda\tau_x}{8}\|\nabla_y f(x_k, y_k)\|^2.$$

Using Lemma A.7 and $\mu_y$-PL condition in $y$ and plugging in $\lambda = 14L^2/\mu_y^2$ yields

$$\frac{7\tau_x}{8}\|\nabla_x f(x_k, y_k) - \nabla g(x_k)\|^2 \leq \frac{\lambda\tau_x}{16}\|\nabla_y f(x_k, y_k)\|^2.$$

Thus,

$$\mathbb{E}[\mathcal{V}_{k+1}] \leq \mathcal{V}_k - \frac{\tau_x}{8}\|\nabla g(x_k)\|^2 - \frac{\lambda\tau_x}{16}\|\nabla_y f(x_k, y_k)\|^2.$$

$\square$

Now it is sufficient to show the convergence of SVRG-GDA.

**Theorem C.1.** *Under Assumption 3.2 and 3.1, if we let $SM = \lceil 8/(\mu_x \tau_x)\rceil$, $T = \lceil \log(1/\epsilon)\rceil$ and $M, B, \tau_x, \tau_y$ defined in Lemma C.1, then the following statement holds true for Algorithm 5:*

$$\mathbb{E}\left[\tilde{\mathcal{A}}_{t+1} + \frac{\lambda\tau_x}{\tau_y}\tilde{\mathcal{B}}_{t+1}\right] \leq \frac{1}{2}\left(\tilde{\mathcal{A}}_t + \frac{\lambda\tau_x}{\tau_y}\tilde{\mathcal{B}}_t\right),$$

*where $\tilde{\mathcal{A}}_t = g(x_t) - g(x^*)$ and $\tilde{\mathcal{B}}_t = g(x_t) - f(x_t, y_t)$. Furthermore, let $\alpha = 2/3$, then it requires $\mathcal{O}((n + n^{2/3}\kappa_x \kappa_y^2)\log(1/\epsilon))$ stochastic first-order calls to achieve $g(x_T) - g(x^*) \leq \epsilon$ in expectation.*

*Proof.* By Lemma C.1 and Lemma A.5 that $g$ satisfies $\mu_x$-PL in $x$ and Assumption 3.2 that function $f$ satisfies $\mu_y$-PL in $y$, we have

$$\mathbb{E}[\mathcal{V}_{s,k+1}] \leq \mathcal{V}_{s,k} - \frac{\mu_x\tau_x}{4}\mathcal{A}_{s,k} - \frac{\mu_y\tau_y}{8}\frac{\lambda\tau_x}{\tau_y}\mathcal{B}_{s,k} \leq \mathcal{V}_{s,k} - \frac{\mu_x\tau_x}{4}\mathcal{V}_{s,k},$$

where in the last inequality we use $\mu_x\tau_x/4 \leq \mu_y\tau_y/8$. Telescoping for $k = 0, 1, \ldots, M - 1$ and $s = 0, 1, \ldots, S - 1$ and rearranging, we can see that in round $t$, it holds that

$$\frac{1}{SM}\sum_{s=0}^{S-1}\sum_{k=0}^{M-1}\mathcal{V}_{s,k} \leq \frac{4}{\mu_x\tau_x SM}(\mathcal{V}_{0,0} - \mathcal{V}_{S,M}) \leq \frac{1}{2}\mathcal{V}_{0,0},$$

where the last inequality is due to the choice of $S$.

The above inequality is exactly equivalent to what we want to prove:

$$\mathbb{E}\left[\tilde{\mathcal{A}}_{t+1} + \frac{\lambda\tau_x}{\tau_y}\tilde{\mathcal{B}}_{t+1}\right] \leq \frac{1}{2}\left(\tilde{\mathcal{A}}_t + \frac{\lambda\tau_x}{\tau_y}\tilde{\mathcal{B}}_t\right).$$

Note that we have $M = \mathcal{O}(n^{3\alpha/2})$ and $S = \mathcal{O}(\kappa_x\kappa_y^2/n^{\alpha/2})$, then the complexity is

$$\mathcal{O}\left((n + SM + Sn)\log\left(\frac{1}{\epsilon}\right)\right) = \mathcal{O}\left((n + (n^\alpha + n^{1-\alpha/2})\kappa_x\kappa_y^2)\log\left(\frac{1}{\epsilon}\right)\right).$$

Plugging in $\alpha = 2/3$ yields the desired complexity and it can also be seen that it is also the best choice of $\alpha$.

$\square$

# D    Proof of Section 4

In this section, we show that why SPIDER type stochastic gradient estimators outperforms SVRG with complete proofs. Using recursive updates by SPIDER, the variance of gradients are under control.

**Lemma D.1** (Fang et al. [12, Modified form Lemma 1]). *In Algorithm 1, it holds true that*

$$\mathbb{E}[\|G_x(x_k, y_k) - \nabla_x f(x_k, y_k)\|^2] \leq \frac{L^2}{B}\sum_{j=(n_k-1)M}^{k}\left(\tau_x^2\mathbb{E}[\|G_x(x_j, y_j)\|^2] + \tau_y^2\mathbb{E}[\|G_y(x_j, y_j)\|^2]\right),$$

$$\mathbb{E}[\|G_y(x_k, y_k) - \nabla_y f(x_k, y_k)\|^2] \leq \frac{L^2}{B}\sum_{j=(n_k-1)M}^{k}\left(\tau_x^2\mathbb{E}[\|G_x(x_j, y_j)\|^2] + \tau_y^2\mathbb{E}[\|G_y(x_j, y_j)\|^2]\right),$$

*where $n_k = \lceil k/M\rceil$ and $(n_k - 1)M \leq k \leq n_k M - 1$.*

The following lemma describes the main convergence property of SPIDER-GDA.

**Lemma D.2.** *Under Assumption 6.1 and 3.1, setting all the parameters as defined in Theorem 4.1, then it holds true that*

$$\mathbb{E}[\|\nabla_x f(\tilde{x}_{t+1}, \tilde{y}_{t+1})\|^2 + \|\lambda\nabla_y f(\tilde{x}_{t+1}, \tilde{y}_{t+1})\|^2] \leq \frac{64}{\tau_x K}\mathbb{E}\left[\tilde{\mathcal{A}}_t + \frac{\lambda\tau_x}{\tau_y}\tilde{\mathcal{B}}_t\right],$$

*where $\tilde{\mathcal{A}}_t = g(\tilde{x}_t) - g(x^*)$ and $\tilde{\mathcal{B}}_t = g(\tilde{x}_t) - f(\tilde{x}_t, \tilde{y}_t)$.*

*Proof.* First of all, we fix $t$ and analyze the inner loop. We define the Lyapunov function as:

$$\mathcal{V}_{t,k} = \mathcal{A}_{t,k} + \frac{\lambda \tau_x}{\tau_y} \mathcal{B}_{t,k},$$

where $\mathcal{A}_{t,k} = g(x_{t,k}) - g(x^*)$ and $\mathcal{B}_{t,k} = g(x_{t,k}) - f(x_{t,k}, y_{t,k})$.

For simplification, we omit the subscripts $t$ when there is no ambiguity. Note that $g(x)$ is $(2L^2/\mu_y)$-smooth, we have

$$\mathbb{E}[g(x_{k+1}) - g(x^*)]$$

$$\leq \mathbb{E}\left[ g(x_k) - g(x^*) + \nabla g(x_k)^\top (x_{k+1} - x_k) + \frac{L^2}{\mu_y} \|x_{k+1} - x_k\|^2 \right]$$

$$= \mathbb{E}\left[ g(x_k) - g(x^*) - \tau_x \nabla g(x_k)^\top G_x(x_k, y_k) + \frac{L^2 \tau_x^2}{\mu_y} \|G_x(x_k, y_k)\|^2 \right]$$

$$= \mathbb{E}\left[ g(x_k) - g(x^*) - \tau_x (\nabla g(x_k) - G_x(x_k, y_k))^\top G_x(x_k, y_k) + \left( \frac{L^2 \tau_x^2}{\mu_y} - \tau_x \right) \|G_x(x_k, y_k)\|^2 \right]$$

$$\leq \mathbb{E}\left[ g(x_k) - g(x^*) + \frac{\tau_x}{2} \|\nabla g(x_k) - G_x(x_k, y_k)\|^2 + \left( \frac{L^2 \tau_x^2}{\mu_y} - \frac{\tau_x}{2} \right) \|G_x(x_k, y_k)\|^2 \right]$$

$$\leq \mathbb{E}\left[ g(x_k) - g(x^*) + \tau_x \|\nabla g(x_k) - \nabla_x f(x_k, y_k)\|^2 \right]$$

$$+ \mathbb{E}\left[ \tau_x \|G_x(x_k, y_k) - \nabla_x f(x_k, y_k)\|^2 - \frac{\tau_x}{4} \|G_x(x_k, y_k)\|^2 \right],$$

$$(27)$$

where the second inequality follows from the fact that $\mathbb{E}[a^\top b] \leq \frac{1}{2}\mathbb{E}[\|a\|^2] + \frac{1}{2}\|b\|^2$; the third inequality is because we have $\tau_x \leq \mu_y/(4L^2)$. Similarly, we can show that

$$\mathbb{E}[f(x_k, y_k) - f(x_{k+1}, y_k)]$$

$$\leq \mathbb{E}\left[ -\nabla_x f(x_k, y_k)^\top (x_{k+1} - x_k) + \frac{L}{2} \|x_{k+1} - x_k\|^2 \right]$$

$$= \mathbb{E}\left[ \tau_x \nabla_x f(x_k, y_k)^\top G_x(x_k, y_k) + \frac{\tau_x^2 L}{2} \|G_x(x_k, y_k)\|^2 \right]$$

$$= \mathbb{E}\left[ \tau_x (\nabla_x f(x_k, y_k) - G_x(x_k, y_k))^\top G_x(x_k, y_k) + \left( \frac{\tau_x^2 L}{2} + \tau_x \right) \|G_x(x_k, y_k)\|^2 \right]$$

$$\leq \mathbb{E}\left[ \frac{\tau_x}{2} \|\nabla_x f(x_k, y_k) - G_x(x_k, y_k)\|^2 + 2\tau_x \|G_x(x_k, y_k)\|^2 \right],$$

and

$$\mathbb{E}[f(x_{k+1}, y_k) - f(x_{k+1}, y_{k+1})]$$

$$\leq \mathbb{E}\left[ -\nabla_y f(x_{k+1}, y_k)^\top (y_{k+1} - y_k) + \frac{L}{2} \|y_{k+1} - y_k\|^2 \right]$$

$$= \mathbb{E}\left[ -\tau_y \nabla_y f(x_{k+1}, y_k)^\top G_y(x_k, y_k) + \frac{\tau_y^2 L}{2} \|G_y(x_k, y_k)\|^2 \right]$$

$$= \mathbb{E}\left[ -\tau_y (\nabla_y f(x_{k+1}, y_k) - G_y(x_k, y_k))^\top G_y(x_k, y_k) + \left( \frac{\tau_y^2 L}{2} - \tau_y \right) \|G_y(x_k, y_k)\|^2 \right]$$

$$\leq \mathbb{E}\left[ \frac{\tau_y}{2} \|\nabla_y f(x_{k+1}, y_k) - G_y(x_k, y_k)\|^2 + \left( \frac{\tau_y^2 L}{2} - \frac{\tau_y}{2} \right) \|G_y(x_k, y_k)\|^2 \right]$$

$$\leq \mathbb{E}\left[ \tau_y \|\nabla_y f(x_k, y_k) - G_y(x_k, y_k)\|^2 + \tau_y \|\nabla_y f(x_k, y_k) - \nabla_y f(x_{k+1}, y_k)\|^2 \right]$$

$$+ \mathbb{E}\left[ \left( \frac{\tau_y^2 L}{2} - \frac{\tau_y}{2} \right) \|G_y(x_k, y_k)\|^2 \right]$$

$$\leq \mathbb{E}\left[ \tau_y \|\nabla_y f(x_k, y_k) - G_y(x_k, y_k)\|^2 + \tau_y \tau_x^2 L^2 \|G_x(x_k, y_k)\|^2 \right]$$

$$+ \mathbb{E}\left[\left(\frac{\tau_y^2 L}{2} - \frac{\tau_y}{2}\right)\|G_y(x_k, y_k)\|^2\right]$$

$$\leq \mathbb{E}\left[\tau_y\|\nabla_y f(x_k, y_k) - G_y(x_k, y_k)\|^2 + \tau_x\|G_x(x_k, y_k)\|^2 - \frac{\tau_y}{4}\|G_y(x_k, y_k)\|^2\right],$$

where we use $\tau_y \leq 1/(2L)$ and $\tau_x \leq 1/L$ and Young's inequality.

Summing up the above two inequalities, we have

$$\mathbb{E}[f(x_k, y_k) - f(x_{k+1}, y_{k+1})] \leq \mathbb{E}\left[\frac{\tau_x}{2}\|\nabla_x f(x_k, y_k) - G_x(x_k, y_k)\|^2 + 3\tau_x\|G_x(x_k, y_k)\|^2\right]$$
$$+ \mathbb{E}\left[\tau_y\|\nabla_y f(x_k, y_k) - G_y(x_k, y_k)\|^2 - \frac{\tau_y}{4}\|G_y(x_k, y_k)\|^2\right].$$

Combing with inequality (27), it can be seen that

$$\mathbb{E}[\mathcal{B}_{k+1}] = \mathbb{E}[g(x_{k+1}) - f(x_{k+1}, y_{k+1})]$$
$$\leq \mathbb{E}[g(x_{k+1}) - g(x_k) + g(x_k) - f(x_k, y_k) + f(x_k, y_k) - f(x_{k+1}, y_{k+1})]$$
$$\leq \mathbb{E}\left[\mathcal{B}_k + \tau_x\|\nabla g(x_k) - \nabla_x f(x_k, y_k)\|^2 + \frac{3\tau_x}{2}\|G_x(x_k, y_k) - \nabla_x f(x_k, y_k)\|^2\right]$$
$$+ \mathbb{E}\left[\tau_y\|G_y(x_k, y_k) - \nabla_y f(x_k, y_k)\|^2 - \frac{\tau_y}{4}\|G_y(x_k, y_k)\|^2 + 3\tau_x\|G_x(x_k, y_k)\|^2\right].$$

Therefore, using $24\lambda\tau_x \leq \tau_y$ and inequality (27) again, we obtain

$$\mathbb{E}[\mathcal{V}_{k+1}] = \mathbb{E}\left[\mathcal{A}_{k+1} + \frac{\lambda\tau_x}{\tau_y}\mathcal{B}_{k+1}\right]$$
$$\leq \mathbb{E}\left[\mathcal{A}_k + \frac{\lambda\tau_x}{\tau_y}\mathcal{B}_k + \left(\tau_x + \frac{\lambda\tau_x}{\tau_y}\right)\|\nabla g(x_k) - \nabla_x f(x_k, y_k)\|^2\right]$$
$$+ \mathbb{E}\left[\left(\tau_x + \frac{3\lambda\tau_x^2}{2\tau_y}\right)\|G_x(x_k, y_k - \nabla_x f(x_k, y_k)\|^2\right]$$
$$+ \mathbb{E}\left[\left(\frac{3\lambda\tau_x^2}{\tau_y} - \frac{\tau_x}{4}\right)\|G_x(x_k, y_k)\|^2\right]$$
$$+ \mathbb{E}\left[\lambda\tau_x\|G_y(x_k, y_k) - \nabla_y f(x_k, y_k)\|^2 - \frac{\lambda\tau_x}{4}\|G_y(x_k, y_k)\|\right]$$
$$\leq \mathbb{E}\left[\mathcal{A}_k + \frac{\lambda\tau_x}{\tau_y}\mathcal{B}_k + 2\tau_x\|\nabla g(x_k) - \nabla_x f(x_k, y_k)\|^2\right]$$
$$+ \mathbb{E}\left[\frac{5\tau_x}{2}\|G_x(x_k, y_k) - \nabla_x f(x_k, y_k)\|^2 - \frac{\tau_x}{8}\|G_x(x_k, y_k)\|^2\right]$$
$$+ \mathbb{E}\left[\lambda\tau_x\|G_y(x_k, y_k) - \nabla_y f(x_k, y_k)\|^2 - \frac{\lambda\tau_x}{4}\|G_y(x_k, y_k)\|^2\right].$$

Furthermore,

$$\mathbb{E}[\mathcal{V}_{k+1}] \leq \mathbb{E}\left[\mathcal{V}_k + \frac{2L^2\tau_x}{\mu_y^2}\|\nabla_y f(x_k, y_k)\|^2\right]$$
$$+ \mathbb{E}\left[\frac{5\tau_x}{2}\|G_x(x_k, y_k) - \nabla_x f(x_k, y_k)\|^2 - \frac{\tau_x}{8}\|G_x(x_k, y_k)\|^2\right]$$
$$+ \mathbb{E}\left[\lambda\tau_x\|G_y(x_k, y_k) - \nabla_y f(x_k, y_k)\|^2 - \frac{\lambda\tau_x}{4}\|G_y(x_k, y_k)\|^2\right]$$
$$\leq \mathbb{E}\left[\mathcal{V}_k + \frac{5\tau_x}{2}\|G_x(x_k, y_k) - \nabla_x f(x_k, y_k)\|^2 - \frac{\tau_x}{8}\|G_x(x_k, y_k)\|^2\right]$$
$$+ \mathbb{E}\left[\frac{9\lambda\tau_x}{8}\|G_y(x_k, y_k) - \nabla_y f(x_k, y_k)\|^2 - \frac{\lambda\tau_x}{8}\|G_y(x_k, y_k)\|^2\right].$$

Above, the first inequality follows from Lemma A.7 and the second inequality uses Young's inequality that $\mathbb{E}[\|a - b\|^2] \leq \mathbb{E}[\|a\|^2 + \|b\|^2]$ and $\lambda = 32L^2/\mu_y^2$.

Now, plug in the variance bound of Spider given by Lemma D.1 and $B = M$, we have

$$
\mathbb{E}[\mathcal{V}_{k+1}] \leq \mathbb{E}\left[\mathcal{V}_k + \left(\frac{5}{2} + \frac{9\lambda}{8}\right)\frac{\tau_x^3 L^2}{M}\sum_{j=(n_k-1)M}^{k}\|G_x(x_j, y_j)\|^2 - \frac{\tau_x}{8}\|G_x(x_k, y_k)\|^2\right]
$$

$$
+ \mathbb{E}\left[\left(\frac{5}{2} + \frac{9\lambda}{8}\right)\frac{\tau_x \tau_y^2 L^2}{M}\sum_{j=(n_k-1)M}^{k}\|G_y(x_j, y_j)\|^2 - \frac{\lambda \tau_x}{8}\|G_y(x_k, y_k)\|^2\right]
$$

$$
\leq \mathbb{E}\left[\mathcal{V}_k + \frac{5\lambda \tau_x^3 L^2}{4M}\sum_{j=(n_k-1)M}^{k}\|G_x(x_j, y_j)\|^2 - \frac{\tau_x}{8}\|G_x(x_k, y_k)\|^2\right]
$$

$$
+ \mathbb{E}\left[\frac{5\lambda \tau_x \tau_y^2 L^2}{4M}\sum_{j=(n_k-1)M}^{k}\|G_y(x_j, y_j)\|^2 - \frac{\lambda \tau_x}{8}\|G_y(x_k, y_k)\|^2\right].
$$

Now we telescope for $i = (n_k - 1)M, \cdots, k$.

$$
\mathbb{E}[\mathcal{V}_{k+1}] \leq \mathbb{E}\left[\mathcal{V}_{(n_k-1)M}\right]
$$

$$
+ \mathbb{E}\left[\sum_{i=(n_k-1)M}^{k}\left(\sum_{j=(n_k-1)M}^{i}\frac{5\lambda \tau_x^3 L^2}{4M}\|G_x(x_j, y_j)\|^2 - \frac{\tau_x}{8}\|G_x(x_i, y_i)\|^2\right)\right]
$$

$$
+ \mathbb{E}\left[\sum_{i=(n_k-1)M}^{k}\left(\sum_{j=(n_k-1)M}^{i}\frac{5\lambda \tau_x \tau_y^2 L^2}{4M}\|G_y(x_j, y_j)\|^2 - \frac{\lambda \tau_x}{8}\|G_y(x_i, y_i)\|^2\right)\right]
$$

$$
\leq \mathbb{E}\left[\mathcal{V}_{(n_k-1)M} + \sum_{j=(n_k-1)M}^{k}\left(\frac{5\lambda \tau_x^3 L^2}{4}\|G_x(x_j, y_j)\|^2 - \frac{\tau_x}{8}\|G_x(x_j, y_j)\|^2\right)\right]
$$

$$
+ \mathbb{E}\left[\sum_{j=(n_k-1)M}^{k}\left(\frac{5\lambda \tau_x \tau_y^2 L^2}{4}\|G_y(x_j, y_j)\|^2 - \frac{\lambda \tau_x}{8}\|G_y(x_j, y_j)\|^2\right)\right]
$$

$$
\leq \mathbb{E}\left[\mathcal{V}_{(n_k-1)M} - \frac{\tau_x}{16}\sum_{j=(n_k-1)M}^{k}\|G_x(x_j, y_j)\|^2 - \frac{\lambda \tau_x}{16}\sum_{j=(n_k-1)M}^{k}\|G_y(x_j, y_j)\|^2\right],
$$

where we use $\lambda \tau_x^2 L^2 \leq 1/20$ and $\tau_y^2 L^2 \leq 1/20$ in the last inequality.

From now on, we need to write down the subscripts with respect to $t$. Telescope for $k = 0, \cdots, K-1$.

$$
\mathbb{E}[\mathcal{V}_{t,K}] \leq \mathbb{E}\left[\mathcal{V}_{t,0} - \frac{\tau_x}{16}\sum_{k=0}^{K-1}\|G_x(x_{t,k}, y_{t,k})\|^2 - \frac{\lambda \tau_x}{16}\sum_{k=0}^{K-1}\|G_y(x_{t,k}, y_{t,k})\|^2\right].
$$

Noting that we choose $(\tilde{x}_{t+1}, \tilde{y}_{t+1})$ from $\{(x_{t,k}, y_{t,k})\}_{k=0}^{K-1}$ uniformly at random, we have

$$
\mathbb{E}\left[\frac{\tau_x}{16}\|G_x(\tilde{x}_{t+1}, \tilde{y}_{t+1})\|^2 + \frac{\lambda \tau_x}{16}\|G_y(\tilde{x}_{t+1}, \tilde{y}_{t+1})\|^2\right] \leq \frac{1}{K}\mathbb{E}\left[\tilde{\mathcal{A}}_t + \frac{\lambda \tau_x}{\tau_y}\tilde{\mathcal{B}}_t\right]. \qquad (28)
$$

Additionally, denote random variable $\xi_t$ as the index from $k = 0, 1, \cdots, K-1$ that is chosen as $(\tilde{x}_{t+1}, \tilde{y}_{t+1})$, it holds that

$$
\mathbb{E}[\|G_x(\tilde{x}_{t+1}, \tilde{y}_{t+1}) - \nabla_x f(\tilde{x}_{t+1}, \tilde{y}_{t+1})\|^2]
$$

$$
\leq \mathbb{E}\left[\frac{L^2 \tau_x^2}{B}\sum_{j=(n_{\xi_t}-1)M}^{\xi_t}\|G_x(x_j, y_j)\|^2\right]
$$

$$\leq \frac{L^2\tau_x^2 M}{TB} \sum_{j=0}^{K-1} \mathbb{E}\|G_x(x_j, y_j)\|^2$$

$$= \frac{L^2\tau_x^2}{K} \sum_{j=0}^{K-1} \mathbb{E}\|G_x(x_j, y_j)\|^2$$

$$= L^2\tau_x^2 \mathbb{E}\|G_x(\tilde{x}_{t+1}, \tilde{y}_{t+1})\|^2,$$

where we use Lemma D.1 again in the first inequality; the second inequality holds because the probability that $n_{\xi_t} = 1, 2, \cdots, n_K$ is less than or equal to $M/K$. Similarly,

$$\mathbb{E}\|G_x(\tilde{x}_{t+1}, \tilde{y}_{t+1}) - \nabla_x f(\tilde{x}_{t+1}, \tilde{y}_{t+1})\|^2 \leq L^2\tau_x^2 \mathbb{E}\|G_x(\tilde{x}_{t+1}, \tilde{y}_{t+1})\|^2.$$

Now it is sufficient to show that

$$\mathbb{E}[\|\nabla_x f(\tilde{x}_{t+1}, \tilde{y}_{t+1})\|^2 + \lambda\|\nabla_y f(\tilde{x}_{t+1}, \tilde{y}_{t+1})\|^2]$$
$$\leq \mathbb{E}[2\|\nabla_x f(\tilde{x}_{t+1}, \tilde{y}_{t+1}) - G_x(\tilde{x}_{t+1}, \tilde{y}_{t+1})\|^2 + 2\|G_x(\tilde{x}_{t+1}, \tilde{y}_{t+1})\|^2]$$
$$\quad + \mathbb{E}[2\|\nabla_y f(\tilde{x}_{t+1}, \tilde{y}_{t+1}) - G_y(\tilde{x}_{t+1}, \tilde{y}_{t+1})\|^2 + 2\|G_y(\tilde{x}_{t+1}, \tilde{y}_{t+1})\|^2]$$
$$\leq 2(1 + L^2\tau_x^2)\mathbb{E}[\|G_x(\tilde{x}_{t+1}, \tilde{y}_{t+1})\|^2] + 2\lambda(1 + L^2\tau_y^2)\mathbb{E}[\|G_y(\tilde{x}_{t+1}, \tilde{y}_{t+1})\|^2]$$
$$\leq 4\mathbb{E}\|G_x(\tilde{x}_{t+1}, \tilde{y}_{t+1})\|^2 + 4\lambda\mathbb{E}\|G_y(\tilde{x}_{t+1}, \tilde{y}_{t+1})\|^2$$
$$\leq \frac{64}{\tau_x K}\mathbb{E}\left[\tilde{\mathcal{A}}_t + \frac{\lambda\tau_x}{\tau_y}\tilde{\mathcal{B}}_t\right],$$

where we use inequality (28) in the last line.

$\square$

Equipped with the above lemma, we can easily prove Theorem 4.1.

### D.1 Proof of Theorem 4.1

*Proof.* Using the properties of PL condition, it holds true that

$$\mathbb{E}\left[\tilde{\mathcal{A}}_{t+1} + \frac{\lambda\tau_x}{\tau_y}\tilde{\mathcal{B}}_{t+1}\right]$$
$$\leq \mathbb{E}\left[\frac{1}{2\mu_x}\|\nabla g(\tilde{x}_{t+1})\|^2 + \frac{\lambda\tau_x}{2\mu_y\tau_y}\|\nabla_y f(\tilde{x}_{t+1}, \tilde{y}_{t+1})\|^2\right]$$
$$\leq \mathbb{E}\left[\frac{1}{2\mu_x}\|\nabla g(\tilde{x}_{t+1})\|^2 + \frac{1}{48\mu_y}\|\nabla_y f(\tilde{x}_{t+1}, \tilde{y}_{t+1})\|^2\right]$$
$$\leq \mathbb{E}\left[\frac{1}{\mu_x}\|\nabla_x f(\tilde{x}_{t+1}, y_{t+1})\|^2 + \frac{1}{\mu_x}\|\nabla_x f(\tilde{x}_{t+1}, \tilde{y}_{t+1}) - \nabla g(\tilde{x}_{t+1})\|^2\right]$$
$$\quad + \mathbb{E}\left[\frac{1}{48\mu_y}\|\nabla_y f(\tilde{x}_{t+1}, \tilde{y}_{t+1})\|^2\right]$$
$$\leq \mathbb{E}\left[\frac{1}{\mu_x}\|\nabla_x f(\tilde{x}_{t+1}, \tilde{y}_{t+1})\|^2 + \left(\frac{1}{48\mu_y} + \frac{L^2}{\mu_x\mu_y^2}\right)\|\nabla_y f(\tilde{x}_{t+1}, \tilde{y}_{t+1})\|^2\right]$$
$$\leq \mathbb{E}\left[\frac{1}{\mu_x}\|\nabla_x f(\tilde{x}_{t+1}, \tilde{y}_{t+1})\|^2 + \frac{33L^2}{32\mu_x\mu_y^2}\|\nabla_y f(\tilde{x}_{t+1}, \tilde{y}_{t+1})\|^2\right]$$
$$\leq \frac{33}{\mu_x}\mathbb{E}\left[\|\nabla_x f(\tilde{x}_{t+1}, \tilde{y}_{t+1})\|^2 + \lambda\|\nabla_y f(\tilde{x}_{t+1}, \tilde{y}_{t+1})\|^2\right]$$
$$\leq \frac{2112}{\mu_x\tau_x K}\mathbb{E}\left[\tilde{\mathcal{A}}_t + \frac{\lambda\tau_x}{\tau_y}\tilde{\mathcal{B}}_t\right]$$
$$\leq \frac{1}{2}\mathbb{E}\left[\tilde{\mathcal{A}}_t + \frac{\lambda\tau_x}{\tau_y}\tilde{\mathcal{B}}_t\right].$$

Above, in the first inequality we use the definition of PL condition; in the the second we plug in $\lambda \tau_x / \tau_y = 1/24$; the third one is due to Young's inequality; the fourth one follows from Lemma A.7; the fifth and sixth ones are both trivial; in the second last one we use Lemma D.2 and in the last one we plug in our choice of $K$. Therefore, to find $\hat{x}$ such that $g(\hat{x}) - g(x^*) \leq \epsilon$ and $g(\hat{x}) - f(\hat{x}, \hat{y}) \leq 24\epsilon$ in expectation, the complexity is

$$\mathcal{O}\left((n + MK)\log\left(\frac{1}{\epsilon}\right)\right) = \mathcal{O}\left((n + K\sqrt{n})\log\left(\frac{1}{\epsilon}\right)\right) = \mathcal{O}\left((n + \sqrt{n}\kappa_x \kappa_y^2)\log\left(\frac{1}{\epsilon}\right)\right)$$

$\square$

## E   Proof of Section 5

In this section, we present the convergence results of AccSPIDER-GDA when $\gamma = 0$, now $F_k$ can be written as:

$$\min_{x \in \mathbb{R}^{d_x}} \max_{y \in \mathbb{R}^{d_y}} F_k(x, y) \triangleq f(x, y) + \frac{\beta}{2}\|x - x_k\|^2.$$

First of all, we take a closer look at $F_k$. The regularization term $\beta$ transforms the condition number of the problem.

**Lemma E.1.** *Given $\beta > L$, the sub-problem $F_k(x, y)$ is $(\beta - L)$-PL in $x$ , $\mu_y$-PL in $y$ and $(\beta + L)$-smooth if $f$ satisfies $L$-smmoth and $\mu_x$-PL in $x$.*

Strong duality also holds for sub-problem $F_k(x, y)$ since its saddle point exist.

**Lemma E.2.** *Under Assumption 6.1 and 5.1, given $\beta > L$, the sub-problem $F_k(x, y)$ has a unique saddle point.*

*Proof.* Denote $G_k(x) \triangleq \max_{y \in \mathbb{R}^{d_y}} F_k(x, y)$. According to Assumption 5.1, the inner problem $\max_{y \in \mathbb{R}^{d_y}} F_k(x, y)$ has a unique solution $y^*(x)$.

Additionally, it is clear that $F_k(x, y)$ is strongly convex in $x$ for $\beta > L$. Hence, we know that $G_k(x)$ is strongly convex since taking the supremum is an operation that preserve (strong) convexity and . In this case, the outer problem $\min_{x \in \mathbb{R}^{d_x}} G_k(x)$ also has a unique solution $x^*$.

Above, the point $(x^*, y^*)$ is a unique global minimax point of $F_k(x, y)$. And the global minimax point of $F_k(x, y)$ is equivalent to a saddle point of $F_k(x, y)$ by Lemma A.1. $\square$

From now on, throughout this section, we always $(\tilde{x}_k, \tilde{y}_k)$ be the saddle point of $F_{k-1}$ for all $k \geq 1$.

Next, we study the error brought by the inexact solution to the sub-problem. The idea is that when we can controls the precision of $F_k$ with a global constant $\delta$, then the algorithm will be close to the exact proximal point algorithm. We omit the notation of expectation when no ambiguity arises.

**Lemma E.3.** *Suppose $x_{k+1}$ satisfies $\mathbb{E}[\|x_{k+1} - \tilde{x}_{k+1}\|^2] \leq \delta$ for any saddle point $(\tilde{x}_{k+1}, \tilde{y}_{k+1})$ of $F_k$, then it holds that*

$$\mathbb{E}\left[F_k(x_{k+1}, \tilde{y}_{k+1}) - F_k(\tilde{x}_{k+1}, \tilde{y}_{k+1})\right] \leq \frac{(\beta + L)^2 \delta}{2(\beta - L)}.$$

*Proof.* Lemma A.2 tells us that $F_k(\tilde{x}_{k+1}, \tilde{y}_{k+1}) = \min_{x \in \mathbb{R}^{d_x}} F_k(x, \tilde{y}_{k+1})$, thus, we have

$$
\begin{aligned}
&\mathbb{E}[F_k(x_{k+1}, \tilde{y}_{k+1}) - F_k(\tilde{x}_{k+1}, \tilde{y}_{k+1})] \\
&= \mathbb{E}[F_k(x_{k+1}, \tilde{y}_{k+1}) - \min_{x \in \mathbb{R}^{d_x}} F_k(x, \tilde{y}_{k+1})] \\
&\leq \frac{1}{2(\beta - L)}\mathbb{E}[\|\nabla_x F_k(x_{k+1}, \tilde{y}_{k+1})\|^2] \\
&= \frac{1}{2(\beta - L)}\mathbb{E}[\|\nabla_x F_k(x_{k+1}, \tilde{y}_{k+1}) - \nabla_x F_k(\tilde{x}_{k+1}, \tilde{y}_{k+1})\|^2]
\end{aligned}
$$

$$\leq \frac{(\beta + L)^2}{2(\beta - L)} \mathbb{E}[\|x_{k+1} - \tilde{x}_{k+1}\|^2]$$

$$\leq \frac{(\beta + L)^2 \delta}{2(\beta - L)},$$

where the first and third inequalities rely on Lemma E.1 that $F_k(x, y)$ is $(\beta - L)$-PL in $x$ and $(\beta + L)$-smooth and the second equality is dependent on the fact that $\nabla_x F_k(\tilde{x}_{k+1}, \tilde{y}_{k+1}) = 0$.

$\square$

We can see that when we can find a $\delta$-saddle point of $F_k$, then we can approximate $g(x)$ well.

**Lemma E.4.** *Suppose $x_{k+1}$ satisfies $\mathbb{E}\|x_{k+1} - \tilde{x}_{k+1}\|^2 \leq \delta$ for any saddle point $(\tilde{x}_{k+1}, \tilde{y}_{k+1})$ of $F_k$, then it holds that*

$$\mathbb{E}|g(x_{k+1}) - g(\tilde{x}_{k+1})| \leq \left( \frac{(\beta + L)^2}{2(\beta - L)} + \frac{\beta}{2} \right) \delta.$$

*Proof.* By definition we know the relationship between $g(x)$ and $F_k(x, y)$ is given by:

$$g(x) = \max_{y \in \mathbb{R}^{d_y}} f(x, y) = F_k(x, \tilde{y}_{k+1}) - \frac{\beta}{2} \|x - x_k\|^2.$$

Thus,

$$\mathbb{E}|g(x_{k+1}) - g(\tilde{x}_{k+1})|$$

$$= \mathbb{E}\left| \left( F_k(x_{k+1}, \tilde{y}_{k+1}) - \frac{\beta}{2} \|x_{k+1} - x_k\|^2 \right) - \left( F_k(\tilde{x}_{k+1}, \tilde{y}_{k+1}) - \frac{\beta}{2} \|\tilde{x}_{k+1} - x_k\|^2 \right) \right|$$

$$\leq \mathbb{E}\left[ F_k(x_{k+1}, \tilde{y}_{k+1}) - F_k(\tilde{x}_{k+1}, \tilde{y}_{k+1}) + \frac{\beta}{2} \|x_{k+1} - \tilde{x}_{k+1}\|^2 \right]$$

$$\leq \left( \frac{(\beta + L)^2}{2(\beta - L)} + \frac{\beta}{2} \right) \delta,$$

where the second inequality follows from the triangle inequality of distance and the third inequality follows from Lemma E.3.

$\square$

When the sub-problem is solved precisely enough, we can show that $g(x)$ decreases in each iteration.

**Lemma E.5.** *Suppose $x_{k+1}$ satisfies $\mathbb{E}\|x_{k+1} - \tilde{x}_{k+1}\|^2 \leq \delta$ for any saddle point $(\tilde{x}_{k+1}, \tilde{y}_{k+1})$ of $F_k$, then it holds that*

$$\mathbb{E}[g(x_{k+1}) - g(x^*)] \leq \mathbb{E}\left[ g(x_k) - g(x^*) - \frac{\beta}{2} \|x_{k+1} - x_k\|^2 + \frac{(\beta + L)^2 \delta}{2(\beta - L)} \right],$$

*Proof.* Consider the following inequalities:

$$\mathbb{E}[g(x_{k+1}) - g(x^*)] = \mathbb{E}\left[ F_k(x_{k+1}, \tilde{y}_{k+1}) - g(x^*) - \frac{\beta}{2} \|x_{k+1} - x_k\|^2 \right]$$

$$\leq \mathbb{E}\left[ F_k(\tilde{x}_{k+1}, \tilde{y}_{k+1}) - g(x^*) - \frac{\beta}{2} \|x_{k+1} - x_k\|^2 + \frac{(\beta + L)^2 \delta}{2(\beta - L)} \right]$$

$$\leq \mathbb{E}\left[ g(x_k) - g(x^*) - \frac{\beta}{2} \|x_{k+1} - x_k\|^2 + \frac{(\beta + L)^2 \delta}{2(\beta - L)} \right],$$

where in the first inequality we use Lemma E.3, the second inequality is because we know it holds that $F_k(\tilde{x}_{k+1}, \tilde{y}_{k+1}) \leq F_k(x_k, \tilde{y}_{k+1}) = g(x_k)$.

$\square$

Now, we consider how $g(x_k)$ converge to $g(x^*)$ when precision $\delta$ is obtained.

**Lemma E.6.** *Suppose $x_{k+1}$ satisfies $\mathbb{E}\|x_{k+1} - \tilde{x}_{k+1}\|^2 \le \delta$ for any saddle point $(\tilde{x}_{k+1}, \tilde{y}_{k+1})$ of $F_k$, then it holds that*

$$\mathbb{E}[g(x_{k+1}) - g(x^*)] \le \mathbb{E}\left[g(x_k) - g(x^*) - \frac{1}{4\beta}\|\nabla g(\tilde{x}_{k+1})\|^2 + \left(\frac{(\beta + L)^2}{2(\beta - L)} + \frac{\beta}{2}\right)\delta\right].$$

*Proof.* The proof is based on Lemma E.5

$$\begin{aligned}
\mathbb{E}[g(x_{k+1}) - g(x^*)] &\le \mathbb{E}\left[g(x_k) - g(x^*) - \frac{\beta}{2}\|x_{k+1} - x_k\|^2 + \frac{(\beta + L)^2\delta}{2(\beta - L)}\right] \\
&\le \mathbb{E}\left[g(x_k) - g(x^*) - \frac{\beta}{4}\|\tilde{x}_{k+1} - x_k\|^2 + \frac{\beta}{2}\|x_{k+1} - \tilde{x}_{k+1}\|^2 + \frac{(\beta + L)^2\delta}{2(\beta - L)}\right] \\
&= \mathbb{E}\left[g(x_k) - g(x^*) - \frac{1}{4\beta}\|\nabla g(\tilde{x}_{k+1})\|^2 + \left(\frac{(\beta + L)^2}{2(\beta - L)} + \frac{\beta}{2}\right)\delta\right],
\end{aligned}$$

where the second inequality relies on the fact that $-\|a - b\|^2 \le \frac{1}{2}\|a\|^2 + \|b\|^2$. In the last equality we use the fact that $(\tilde{x}_{k+1}, \tilde{y}_{k+1})$ is also a stationary point by Lemma A.1, which implies that $\nabla g(\tilde{x}_{k+1}) + \beta(\tilde{x}_{k+1} - x_k) = 0$.

$\square$

## E.1 Proof of Lemma 5.1

*Proof.* Noting that $g(x)$ satisfies $\mu_x$-PL by Lemma A.5 and using the result of Lemma E.6, we can see that

$$\begin{aligned}
\mathbb{E}[g(x_{k+1}) - g(x^*)] &\le \mathbb{E}\left[g(x_k) - g(x^*) - \frac{1}{4\beta}\|\nabla g(\tilde{x}_{k+1})\|^2 + \left(\frac{(\beta + L)^2}{2(\beta - L)} + \frac{\beta}{2}\right)\delta\right] \\
&\le \mathbb{E}\left[g(x_k) - g(x^*) - \frac{\mu_x}{2\beta}(g(\tilde{x}_{k+1}) - g(x^*)) + \left(\frac{(\beta + L)^2}{2(\beta - L)} + \frac{\beta}{2}\right)\delta\right].
\end{aligned}$$

Using Lemma E.4, we obtain

$$\mathbb{E}[g(x_{k+1}) - g(x^*)] \le \mathbb{E}\left[g(x_k) - g(x^*) - \frac{\mu_x}{2\beta}(g(x_{k+1}) - g(x^*)) + \left(1 + \frac{\mu_x}{2\beta}\right)\left(\frac{(\beta + L)^2}{2(\beta - L)} + \frac{\beta}{2}\right)\delta\right].$$

Rearranging,

$$\mathbb{E}[g(x_{k+1}) - g(x^*)] \le \mathbb{E}\left[\left(1 - \frac{\mu_x}{2\beta + \mu_x}\right)(g(x_k) - g(x^*)) + \left(\frac{(\beta + L)^2}{2(\beta - L)} + \frac{\beta}{2}\right)\delta\right].$$

Let $q \triangleq \mu_x/(2\beta + \mu_x)$ and telescope, then we can obtain that

$$\begin{aligned}
\mathbb{E}[g(x_k) - g(x^*)] &\le (1 - q)^k(g(x_0) - g(x^*)) + \left(\frac{(\beta + L)^2}{2(\beta - L)} + \frac{\beta}{2}\right)\delta\sum_{i=0}^{k-1}(1 - q)^i \\
&\le (1 - q)^k(g(x_0) - g(x^*)) + \left(\frac{(\beta + L)^2}{2(\beta - L)} + \frac{\beta}{2}\right)\frac{\delta}{q}.
\end{aligned}$$

Plugging in $\beta, \delta$ yields the desired statement, $\square$

Now we show that how we can control the precision of the sub-problem $\delta$ recursively to satisfy the condition of Lemma 5.1 that $\mathbb{E}[\|x_k - \tilde{x}_k\|^2 + \|y_k - \tilde{y}_k\|^2] \le \delta$ holds for all $k \ge 1$.

Before that, we need the following lemma showing that when $\|x_k - x_{k+1}\|$ is small, then the distance between the saddle points of $F_k$ and $F_{k+1}$ will be also small. Denote $(\tilde{x}_k, \tilde{y}_k)$ be a saddle point of sub-problem $F_{k-1}$ and $(\tilde{x}_{k+1}, \tilde{y}_{k+1})$ be a saddle point of sub-problem $F_k$.

**Lemma E.7.** *If we let $\beta > L$, then it holds true that*

$$\|\tilde{x}_{k+1} - \tilde{x}_k\|^2 + \|\tilde{y}_{k+1} - \tilde{y}_k\|^2 \le \frac{4\beta^2}{(\beta - L)\mu_y}\|x_k - x_{k-1}\|^2.$$

*Proof.* Noting that $F_{k-1}$ is $(\beta - L)$-PL in $x$ and $\mu_y$-PL in $y$ by Lemma E.1 and using the quadratic growth condition by Lemma A.4, we have

$$\frac{\beta - L}{2}\|\tilde{x}_{k+1} - \tilde{x}_k\|^2 \leq F_{k-1}(\tilde{x}_{k+1}, \tilde{y}_k) - \min_{x \in \mathbb{R}^{d_x}} F_{k-1}(x, \tilde{y}_k) = F_{k-1}(\tilde{x}_{k+1}, \tilde{y}_k) - F_{k-1}(\tilde{x}_k, \tilde{y}_k),$$

$$\frac{\mu_y}{2}\|\tilde{y}_{k+1} - \tilde{y}_k\|^2 \leq \max_{y \in \mathbb{R}^{d_y}} F_{k-1}(\tilde{x}_k, y) - F_{k-1}(\tilde{x}_k, \tilde{y}_{k+1}) = F_{k-1}(\tilde{x}_k, \tilde{y}_k) - F_{k-1}(\tilde{x}_k, \tilde{y}_{k+1}).$$

Combining the above two inequalities, we can see that

$$\frac{\beta - L}{2}\|\tilde{x}_{k+1} - \tilde{x}_k\|^2 + \frac{\mu_y}{2}\|\tilde{y}_{k+1} - \tilde{y}_k\|^2$$

$$\leq F_{k-1}(\tilde{x}_{k+1}, \tilde{y}_k) - F_{k-1}(\tilde{x}_k, \tilde{y}_{k+1})$$

$$= F_k(\tilde{x}_{k+1}, \tilde{y}_k) + \frac{\beta}{2}\|\tilde{x}_{k+1} - x_{k-1}\|^2 - \frac{\beta}{2}\|\tilde{x}_{k+1} - x_k\|^2$$

$$\quad - F_k(\tilde{x}_k, \tilde{y}_{k+1}) - \frac{\beta}{2}\|\tilde{x}_k - x_{k-1}\|^2 + \frac{\beta}{2}\|\tilde{x}_k - x_k\|^2$$

$$\leq \frac{\beta}{2}\|\tilde{x}_{k+1} - x_{k-1}\|^2 - \frac{\beta}{2}\|\tilde{x}_{k+1} - x_k\|^2 - \frac{\beta}{2}\|\tilde{x}_k - x_{k-1}\|^2 + \frac{\beta}{2}\|\tilde{x}_k - x_k\|^2$$

$$= \beta(\tilde{x}_{k+1} - \tilde{x}_k)^\top (x_k - x_{k-1})$$

$$\leq \frac{\beta - L}{4}\|\tilde{x}_{k+1} - \tilde{x}_k\|^2 + \frac{\beta^2}{\beta - L}\|x_k - x_{k-1}\|^2,$$

where the second inequality is based on $F_k(\tilde{x}_{k+1}, \tilde{y}_k) \leq F_k(\tilde{x}_{k+1}, \tilde{y}_{k+1}) \leq F_k(\tilde{x}_k, \tilde{y}_{k+1})$ by $(\tilde{x}_{k+1}, \tilde{y}_{k+1})$ is a saddle point of $F_k$. In the last inequality we use Young's inequality. Rearranging,

$$\frac{\beta - L}{4}\|\tilde{x}_{k+1} - \tilde{x}_k\|^2 + \frac{\mu_y}{2}\|\tilde{y}_{k+1} - \tilde{y}_k\|^2 \leq \frac{\beta^2}{\beta - L}\|x_k - x_{k-1}\|^2.$$

Since we have $(\beta - L)/\mu_y \geq L/4 \geq \mu_y/4$, we can obtain that

$$\|\tilde{x}_{k+1} - \tilde{x}_k\|^2 + \|\tilde{y}_{k+1} - \tilde{y}_k\|^2 \leq \frac{4\beta^2}{(\beta - L)\mu_y}\|x_k - x_{k-1}\|^2,$$

$\square$

An additional bound is for the use of the base case, i.e. $k = 0$.

**Lemma E.8.** *If we let $\beta > L$, then it holds true that*

$$\|x_0 - \tilde{x}_1\|^2 + \|y_0 - \tilde{y}_1\|^2 \leq \frac{2}{\mu_y}(g(x_0) - g(x^*))$$

*Proof.* Note that $F_1$ satisfies $(\beta - L)$-PL in $x$ and $\mu_y$-PL in $y$. We can bound $\|x_0 - \tilde{x}_1\|^2 + \|y_0 - \tilde{y}_1\|^2$ as follows:

$$\frac{\beta - L}{2}\|x_0 - \tilde{x}_1\|^2 \leq F_0(x_0, \tilde{y}_1) - \min_{x \in \mathbb{R}^{d_x}} F_0(x, \tilde{y}_1) = F_0(x_0, \tilde{x}_1) - F_0(\tilde{x}_1, \tilde{y}_1),$$

$$\frac{\mu_y}{2}\|y_0 - \tilde{y}_1\|^2 \leq \max_{y \in \mathbb{R}^{d_y}} F_0(\tilde{x}_1, y) - F_0(\tilde{x}_1, y_0) = F_0(\tilde{x}_1, \tilde{y}_1) - F_0(\tilde{x}_1, y_0).$$

Combining the above two inequalities, we have

$$\frac{\beta - L}{2}\|x_0 - \tilde{x}_1\|^2 + \frac{\mu_y}{2}\|y_0 - \tilde{y}_1\|^2$$

$$\leq F_0(x_0, \tilde{y}_1) - F_0(\tilde{x}_1, y_0)$$

$$= f(x_0, \tilde{y}_1) - f(\tilde{x}_1, y_0) - \frac{\beta}{2}\|x_0 - \tilde{x}_1\|^2$$

$$\leq f(x_0, \tilde{y}_1) - f(\tilde{x}_1, y_0)$$

$$\leq \max_{y \in \mathbb{R}^{d_y}} f(x_0, y) - \min_{x \in \mathbb{R}^{d_x}} f(x, y_0)$$

$$= \max_{y \in \mathbb{R}^{d_y}} f(x_0, y) - \min_{x \in \mathbb{R}^{d_x}} \max_{y \in \mathbb{R}^{d_y}} f(x, y) + \min_{x \in \mathbb{R}^{d_x}} \max_{y \in \mathbb{R}^{d_y}} f(x, y) - \min_{x \in \mathbb{R}^{d_x}} f(x, y_0)$$

$$\leq g(x_0) - g(x^*),$$

where we use the definition of $g$ and the fact that $\min_{x \in \mathbb{R}^{d_x}} \max_{y \in \mathbb{R}^{d_y}} f(x, y) \geq \min_{x \in \mathbb{R}^{d_x}} f(x, y_0)$ in the last inequality. Rearranging and noting that $\beta - L \geq L \geq \mu_y$, we finish the proof. $\qquad\square$

## E.2 The Proof of Lemma 5.2

*Proof.* Recall we solve the sub-problem:

$$\max_{y \in \mathbb{R}^{d_y}} \min_{x \in \mathbb{R}^{d_x}} F_k(x, y) = - \min_{x \in \mathbb{R}^{d_x}} \max_{y \in \mathbb{R}^{d_y}} \{-F_k(x, y)\}.$$

It is $\mu_y$-PL in $y$ and $L$-strongly-convex in $x$ and thus clearly satisfies $L$-PL in $x$.

We define $H_k(y) = \min_{x \in \mathbb{R}_{d_x}} F_k(x, y)$. By Lemma A.8 and A.5, we know that $H_k(y)$ is also $\mu_y$-PL in $y$ and it is $12L$-smooth since $F_k$ is $3L$-smooth.

According to Theorem 4.1, We know that SPIDER-GDA makes sure

$$\underbrace{\mathbb{E}\left[H_{k+1}(\tilde{y}_{k+1}) - H_{k+1}(y_{k+1}) + \frac{1}{24}\left(F_{k+1}(x_{k+1}, y_{k+1}) - H_{k+1}(y_{k+1})\right)\right]}_{\text{LHS}}$$

$$\leq \delta_k \underbrace{\mathbb{E}\left[H_{k+1}(\tilde{y}_{k+1}) - H_{k+1}(y_k) + \frac{1}{24}\left(F_{k+1}(x_k, y_k) - H_{k+1}(y_k)\right)\right]}_{\text{RHS}}.$$

For the left hand side (LHS), we bound it according to

$$\mathbb{E}\|y_{k+1} - \tilde{y}_{k+1}\|^2 \leq \frac{2}{\mu_y}\mathbb{E}[H_{k+1}(\tilde{y}_{k+1}) - H_{k+1}(y_{k+1})] \tag{29}$$

(where we use the $\mu_y$-PL condition in $y$) and

$$\begin{aligned}
&\mathbb{E}\|x_{k+1} - \tilde{x}_{k+1}\|^2 \\
&\leq 2\mathbb{E}[\|x_{k+1} - x^*(y_{k+1})\|^2 + \|x^*(y_{k+1}) - \tilde{x}_{k+1}\|^2] \\
&= 2\mathbb{E}[\|x_{k+1} - x^*(y_{k+1})\|^2 + \|x^*(y_{k+1}) - x^*(\tilde{y}_{k+1})\|^2] \\
&\leq 2\mathbb{E}[\|x_{k+1} - x^*(y_{k+1})\|^2 + 18\mathbb{E}[\|y_{k+1} - \tilde{y}_{k+1}\|^2] \\
&\leq \frac{4}{L}\mathbb{E}[F_{k+1}(x_{k+1}, y_{k+1}) - H_{k+1}(y_{k+1})] + \frac{36}{\mu_y}\mathbb{E}[H_{k+1}(\tilde{y}_{k+1}) - H_{k+1}(y_{k+1}))],
\end{aligned} \tag{30}$$

where we use Lemma A.9 in the second last inequality and the PL condition of $F_k(x, y)$ and $H_k(y)$ in the last one. Summing up (29) and (30), we have

$$\begin{aligned}
&\mathbb{E}[\|y_{k+1} - \tilde{y}_{k+1}\|^2 + \|x_{k+1} - \tilde{x}_{k+1}\|^2] \\
&\leq \frac{4}{L}\mathbb{E}[F_{k+1}(x_{k+1}, y_{k+1}) - H_{k+1}(y_{k+1})] + \frac{38}{\mu_y}\mathbb{E}[H_{k+1}(\tilde{y}_{k+1}) - H_{k+1}(y_{k+1}))] \\
&\leq \frac{96}{\mu_y} \times \text{LHS} \\
&\leq \frac{96\delta_k}{\mu_y} \times \text{RHS}.
\end{aligned}$$

For the right hand side (RHS), we bound it using

$$\begin{aligned}
\text{RHS} &= H_{k+1}(\tilde{y}_{k+1}) - H_{k+1}(y_k) + \frac{1}{24}\left(F_{k+1}(x_k, y_k) - H_{k+1}(y_k)\right) \\
&\leq \frac{1}{2\mu_y}\|\nabla H_{k+1}(y_k)\|^2 + \frac{1}{48L}\|\nabla_x F_{k+1}(x_k, y_k)\|^2
\end{aligned}$$

$$= \frac{1}{2\mu_y}\|\nabla H_{k+1}(y_k) - \nabla H_{k+1}(\tilde{y}_{k+1})\|^2 + \frac{1}{48L}\|\nabla_x F_{k+1}(x_k, y_k) - \nabla_x F_{k+1}(x^*(y_k), y_k)\|^2$$

$$\leq \frac{72L^2}{\mu_y}\|y_k - \tilde{y}_{k+1}\|^2 + \frac{3L}{16}\|x_k - x^*(y_k)\|^2$$

$$\leq \frac{72L^2}{\mu_y}\|y_k - \tilde{y}_{k+1}\|^2 + \frac{3L}{8}\|x_k - \tilde{x}_{k+1}\|^2 + \frac{3L}{8}\|x^*(\tilde{y}_{k+1}) - x^*(y_k)\|^2$$

$$\leq \frac{72L^2}{\mu_y}\|y_k - \tilde{y}_{k+1}\|^2 + \frac{L}{24}\|x_k - \tilde{x}_{k+1}\|^2 + \frac{27L}{8}\|y_k - \tilde{y}_{k+1}\|^2.$$

Above, the first inequality is due to the PL condition of $F_k(x, y)$ and $H_k(y)$; the second inequality follows from $H_k(y)$ is $12L$-smooth and $F_k(x, y)$ is $3L$-smooth; the third one directly follows from the Young's inequality; the fourth inequality uses Lemma A.9 and the fact that $F_k(x, y)$ is $L$-PL in $x$ and $3L$-smooth.

Therefore, we obtain

$$\mathbb{E}[\|y_{k+1} - \tilde{y}_{k+1}\|^2 + \|x_{k+1} - \tilde{x}_{k+1}\|^2] \leq \delta_k'(\|y_k - \tilde{y}_{k+1}\|^2 + \|x_k - \tilde{x}_{k+1}\|^2),$$

where

$$\delta_k' = 7236\kappa_y^2\delta_k. \tag{31}$$

$\square$

Now it is sufficient to control $\delta$ recursively.

**Lemma E.9.** *If we solve each sub-problem $F_k$ with precision $\delta_k$ as defined in Theorem 5.1, then for all $k$ it holds true that*

$$\mathbb{E}\|x_k - \tilde{x}_k\|^2 + \|y_k - \tilde{y}_k\|^2 \leq \delta.$$

*Proof.* We prove by induction. Suppose we the following statement holds true for all $1 \leq k' \leq k$ that we have

$$\mathbb{E}\|x_{k'} - \tilde{x}_{k'}\|^2 + \|y_{k'} - \tilde{y}_{k'}\|^2 \leq \delta,$$

Then, by Lemma 5.2 we have

$$\mathbb{E}[\|x_{k+1} - \tilde{x}_{k+1}\|^2 + \|y_{k+1} - \tilde{y}_{k+1}\|^2]$$
$$\leq \delta_k'(\|x_k - \tilde{x}_{k+1}\|^2 + \|y_k - \tilde{y}_{k+1}\|^2)$$
$$\leq 2\delta_k'(\|x_k - \tilde{x}_k\|^2 + \|y_k - \tilde{y}_k\|^2) + 2\delta_k'(\|\tilde{x}_{k+1} - \tilde{x}_k\|^2 + \|\tilde{y}_{k+1} - \tilde{y}_k\|^2)$$
$$\leq 2\delta_k'\delta + \frac{8\beta^2\delta_k'}{(\beta - L)\mu_y}\|x_k - x_{k-1}\|^2,$$

where $\delta_k'$ follows from (31) and we use the induction hypothesis and Lemma E.7 in the third inequality. Note that our choice of $\delta_k$ and the relationship between $\delta_k$ and $\delta_k'$ satisfy

$$\max\left\{2\delta_k'\delta, \frac{8\beta^2\delta_k'\|x_k - x_{k-1}\|^2}{(\beta - L)\mu_y}\right\} \leq \frac{\delta}{2}.$$

Therefore, we can see that

$$\mathbb{E}[\|x_{k+1} - \tilde{x}_{k+1}\|^2 + \|y_{k+1} - \tilde{y}_{k+1}\|^2] \leq \delta,$$

which completes the induction from $k$ to $k + 1$. For the induction base, using Lemma E.8 we have

$$\mathbb{E}[\|x_1 - \tilde{x}_1\|^2 + \|y_1 - \tilde{y}_1\|^2] \leq \delta_0'(\|x_0 - \tilde{x}_1\|^2 + \|y_0 - \tilde{y}_1\|^2)$$
$$\leq \frac{2\delta_0'}{\mu_y}(g(x_0) - g^*)$$
$$\leq \delta.$$

$\square$

## E.3 Proof of Theorem 5.1

Combing Lemma 5.1, Lemma 5.2 and Lemma E.9, we can easily prove Theorem 5.1.

*Proof.* Note that each sub-problem $F_k$ is $3L$-smooth, $L$-PL in $x$ and $\mu_y$-PL in $y$ for $\beta = 2L$. Now if we choose

$$K = \lceil ((2\beta + \mu_x)/\mu_x) \log(2/\epsilon) \rceil = \mathcal{O}(\kappa_x \log(1/\epsilon)),$$

then by Lemma 5.1 it is sufficient to guarantee that $\mathbb{E}[g(x_K) - g(x^*)] \leq \epsilon$, while solving each sub-problem $F_k$ requires no more than $T_k \leq a(n + \sqrt{n}\kappa_y) \log(\kappa_y/\delta_k)$ first-order oracle calls in expectation by Lemma 5.2, where $a$ is an independent positive constant.

Now we telescope the inequality in Lemma E.5 and we can obtain

$$\sum_{k=0}^{K-1} \mathbb{E}\left[\|x_{k+1} - x_k\|^2\right] \leq \frac{2}{\beta}(g(x_0) - g^*) + \frac{(\beta + L)^2\delta}{\beta(\beta - L)}. \tag{32}$$

Note that we have

$$\frac{1}{\delta_k} \leq \omega \times \max\left\{4, \frac{16\kappa_y\|x_k - x_{k-1}\|^2}{\delta}\right\} \leq \omega \times \left(4 + \frac{16\kappa_y\|x_k - x_{k-1}\|^2}{\delta}\right),$$

by the choice of $\delta_k$ for all $k \geq 1$ (5), where $\omega = 7236\kappa_y^2$. Denote $C = a(n + \sqrt{n}\kappa)$, then

$$
\begin{aligned}
&\sum_{k=0}^{K} T_k \\
&= T_0 + \sum_{k=1}^{K} T_k \\
&\leq C \log\left(\omega \times \frac{2\kappa_y(g(x_0) - g^*)}{\delta\mu_y}\right) + C\sum_{k=1}^{K} \log\left(\frac{\kappa_y}{\delta_k}\right) \\
&\leq C \log\left(\omega \times \frac{2\kappa_y(g(x_0) - g^*)}{\delta\mu_y}\right) + C\sum_{k=1}^{K} \log\left(\omega \times \left(4\kappa_y + \frac{16\kappa_y^2\|x_k - x_{k-1}\|^2}{\delta}\right)\right) \\
&\leq C \log\left(\omega \times \frac{2\kappa_y(g(x_0) - g^*)}{\delta\mu_y}\right) + CK \log\left(\omega \times \sum_{k=1}^{K} \left(4\kappa_y + \frac{16\kappa_y^2\|x_k - x_{k-1}\|^2}{\delta}\right)\right) \\
&= C \log\left(\omega \times \frac{2\kappa_y(g(x_0) - g^*)}{\delta\mu_y}\right) + CK \log\left(\omega \times \sum_{k=0}^{K-1} \left(4\kappa_y + \frac{16\kappa_y^2\|x_k - x_{k-1}\|^2}{\delta}\right)\right).
\end{aligned}
\tag{33}
$$

Above, the second inequality relies on the choice of $\delta_k$ and the third inequality is due to the fact that $(\prod_{i=1}^{n} x_i)^{1/n} \leq \frac{1}{n}\sum_{i=1}^{n} x_i$, which implies that $\sum_{i=1}^{n} \log x_i \leq n \log\left(\frac{1}{n}\sum_{i=1}^{n} x_i\right)$.

Lastly, we use (32) and notice that $\delta$ (3) is dependent on $\epsilon$, $\kappa_x$ and $\omega$ is dependent on $\kappa_y$ to show the SFO complexity of the order

$$\mathcal{O}((n\kappa_x + \sqrt{n}\kappa_x\kappa_y) \log(1/\epsilon) \log(\kappa_x\kappa_y/\epsilon)).$$

$\square$

## F    Proof of Section 6

First of all, we show the convergence of SVRG-GDA and SPIDER-GDA under one-sided PL condition as studied in Section 6. We reuse the lemmas under two-sided PL condition. It is worth noticing that we can discard the outermost loop with respect to restart strategy for both SVRG-GDA and SPIDER-GDA, i.e we set $T = 1$ in this setting.

## F.1 SVRG-GDA under one-sided PL condition

For SVRG-GDA, we have the following theorem.

**Theorem F.1.** *Under Assumption 6.1 and 3.1, let $T = 1$ and $M, \tau_x, \tau_y, \lambda$ defined in Lemma C.1; $\alpha = 2/3$, $SM = \lceil 8/(\tau_x \epsilon^2) \rceil$. Algorithm 2 can guarantee the output $\hat{x}$ to satisfy $\|\nabla g(\hat{x})\|^2 \leq \epsilon$ in expectation with no more than $\mathcal{O}(n + n^{2/3}\kappa_y^2 L \epsilon^{-2})$ stochastic first-order oracle calls.*

*Proof.* Telescoping for $k = 0, \ldots, M-1$ and $s = 0, \ldots, S-1$ for the inequality in Lemma C.1:

$$\frac{1}{SM} \sum_{s=0}^{S-1} \sum_{k=0}^{M-1} \mathbb{E}[\|\nabla g(x_{s,k})\|^2] \leq \frac{8\mathcal{V}_{0,0}}{\tau_x SM}.$$

Note that we have $M = \mathcal{O}(n^{3\alpha/2})$ and if we let $SM = \lceil 8/(\tau_x \epsilon^2) \rceil$, then $S = \mathcal{O}(L\kappa_y^2/(n^{\alpha/2}\epsilon^2))$, so the complexity is

$$\mathcal{O}(n + SM + Sn) = \mathcal{O}\left(n + \frac{\kappa_y^2 L(n^\alpha + n^{1-\alpha/2})}{\epsilon^2}\right).$$

Plugging in $\alpha = 2/3$ yields the desired complexity. □

## F.2 Proof of Theorem 6.1

Similarly to SVRG-GDA, we can also analyze the convergence of SPIDER-GDA.

*Proof.* It is almost direct result of Lemma D.2. Denote $(\hat{x}, \hat{y})$ the output, then

$$\begin{aligned}
\mathbb{E}\|\nabla g(\hat{x})\|^2 &\leq 2\mathbb{E}\left[\|\nabla_x f(\hat{x}, \hat{y})\|^2 + \|\nabla_x f(\hat{x}, \hat{y}) - \nabla g(\hat{x})\|^2\right] \\
&\leq 2\mathbb{E}\left[\|\nabla_x f(\hat{x}, \hat{y})\|^2 + \frac{L^2}{\mu_y^2}\|\nabla_y f(\hat{x}, \hat{y})\|^2\right] \\
&\leq 2\mathbb{E}[\|\nabla_x f(\hat{x}, \hat{y})\|^2 + \lambda\|\nabla_y f(\hat{x}, \hat{y})\|^2] \\
&\leq \frac{32}{\tau_x K}\mathbb{E}\left[\tilde{\mathcal{A}}_0 + \tilde{\mathcal{B}}_0\right],
\end{aligned}$$

where the first inequality follows from Young's inequality; the second one relies on Lemma A.7; the third one uses the definition of $\lambda$ and the last one uses Lemma D.2.

Since $\tau_x = \mathcal{O}(1/(\kappa_y^2 L))$ and $M = B = \sqrt{n}$, the complexity becomes:

$$\mathcal{O}\left(n + \frac{\sqrt{n}}{\tau_x \epsilon^2}\right) = \mathcal{O}\left(n + \frac{\sqrt{n}\kappa_y^2 L}{\epsilon^2}\right).$$

□

Above, we have show that the complexity of SVRG-GDA is $\mathcal{O}(n + n^{2/3}\kappa_y^2 L\epsilon^{-2})$ and the complexity of SPIDER-GDA is $\mathcal{O}(n + \sqrt{n}\kappa_y^2 L\epsilon^{-2})$ [6]. Thus, we can come to the conclusion that SPIDER-GDA strictly outperforms SVRG-GDA under both two-sided and one-sided PL conditions. In the rest of this section, we mainly focus on the complexity of AccSPIDER-GDA under one-sided PL condition.

In the following lemma, we show that AccSPIDER-GDA converge when we can control the precision of solving each sub-problem with a global constant $\delta$.

---

[6]To be more precise, our theorem only suits the case when $1/(\kappa_y^2 L\epsilon^2) > \sqrt{n}$ for SPIDER-GDA. If not, we can directly set $K = 2M$ to achieve the same convergence result.

### F.3 Proof of Lemma 6.1

*Proof.* Similar to the proof under two-sided PL condition, we begin our proof with Lemma E.6. We can see that

$$\mathbb{E}[g(x_{k+1})] \leq \mathbb{E}\left[g(x_k) - \frac{1}{4\beta}\|\nabla g(\tilde{x}_{k+1})\|^2 + \left(\frac{(\beta+L)^2}{2(\beta-L)} + \frac{\beta}{2}\right)\delta\right]$$

$$\leq \mathbb{E}\left[g(x_k) - \frac{1}{8\beta}\|\nabla g(x_{k+1})\|^2\right]$$

$$+ \mathbb{E}\left[\frac{1}{4\beta}\|\nabla g(\tilde{x}_{k+1}) - \nabla g(x_{k+1})\|^2 + \left(\frac{(\beta+L)^2}{2(\beta-L)} + \frac{\beta}{2}\right)\delta\right]$$

$$\leq \mathbb{E}\left[g(x_k) - \frac{1}{8\beta}\|\nabla g(x_{k+1})\|^2\right]$$

$$+ \mathbb{E}\left[\frac{L^2}{2\mu_y\beta}\|\tilde{x}_{k+1} - x_{k+1}\|^2 + \left(\frac{(\beta+L)^2}{2(\beta-L)} + \frac{\beta}{2}\right)\delta\right]$$

$$\leq \mathbb{E}\left[g(x_k) - \frac{1}{8\beta}\|\nabla g(x_{k+1})\|^2 + \left(\frac{L^2}{2\mu_y\beta} + \frac{(\beta+L)^2}{2(\beta-L)} + \frac{\beta}{2}\right)\delta\right],$$

where we use the fact that $-\|a-b\|^2 \leq \frac{1}{2}\|a\|^2 + \|b\|^2$ in the second inequality, $g(x)$ is $(2L^2/\mu_y)$-smooth in the third one and $\|\tilde{x}_{k+1} - x_{k+1}\|^2 \leq \delta$ in the last one. Telescoping for $k = 0, 1, 2, ...K-1$, we can see that

$$\frac{1}{8\beta}\sum_{k=0}^{K-1}\mathbb{E}\left[\|\nabla g(x_k)\|^2\right] \leq \mathbb{E}\left[g(x_0) - g(x_K) + \left(\frac{L^2}{2\mu_y\beta} + \frac{(\beta+L)^2}{2(\beta-L)} + \frac{\beta}{2}\right)K\delta\right]$$

$$\leq \mathbb{E}\left[g(x_0) - g^* + \left(\frac{L^2}{2\mu_y\beta} + \frac{(\beta+L)^2}{2(\beta-L)} + \frac{\beta}{2}\right)K\delta\right].$$

Divide both sides by $K$, then

$$\frac{1}{K}\sum_{k=0}^{K-1}\mathbb{E}\left[\|\nabla g(x_k)\|^2\right] \leq \mathbb{E}\left[\frac{8\beta(g(x_0) - g^*)}{K} + 8\beta\left(\frac{L^2}{2\mu_y\beta} + \frac{(\beta+L)^2}{2(\beta-L)} + \frac{\beta}{2}\right)\delta\right].$$

Plugging the choice of $\delta$ yields the desired inequality. □

### F.4 Proof of Theorem 6.2

Combing Lemma 6.1, Lemma 5.2 and Lemma E.9, we can easily prove Theorem 6.2. We remark that both the proof Lemma 5.2 and Lemma E.9 only uses the PL property in the direction of $y$, so they can both be directly applied to the one-sided PL case.

*Proof.* Note that each sub-problem $F_k$ is $3L$-smooth, $L$-PL in $x$ and $\mu_y$-PL in $y$ for $\beta = 2L$. Now if we choose

$$K = \lceil 16\beta(g(x_0) - g^*)/\epsilon^2 \rceil = \mathcal{O}(L\epsilon^{-2}),$$

then by Lemma 6.1 it is sufficient to guarantee that $\mathbb{E}[\|g(\hat{x})\|] \leq \epsilon$, while solving each sub-problem $F_k$ requires no more than $T_k \leq a(n + \sqrt{n}\kappa_y)\log(\kappa_y/\delta_k)$ first-order oracle calls in expectation by Lemma 5.2, where $a$ is an independent positive constant.

Therefore, using (32), (33) and noticing that $\delta$ (6) is dependent on $\epsilon$, $\kappa_y$ and $\omega$ in (33) is dependent on $\kappa_y$ to show the SFO complexity of the order

$$\mathcal{O}((n + \sqrt{n}\kappa_y)L\epsilon^{-2}\log(\kappa_y/\epsilon)).$$

□