# OpenReview forum: "Faster Stochastic Algorithms for Minimax Optimization under Polyak-{\L}ojasiewicz Condition"
_NeurIPS.cc/2022/Conference — NeurIPS 2022 Accept_

### Official Review · Reviewer_vsqY · 2022-06-26

**Rating:** 6
**Confidence:** 2
**Soundness:** 3 good
**Presentation:** 3 good
**Contribution:** 3 good

**Summary:**

This work studies a new algorithm for finite sum smooth minimax optimization which has guarantees for a certain class of nonconvex-nonconcave minimax problems, which is formalized by a Polyak-Lojasiewicz (PL) condition. This problem has been previously studied by [42], and the PL condition provides a broad class of optimization problems which are of deep interest to machine learning, including, e.g., deep AUC maximization. The new algorithm shows that a technique related to SPIDER, which reduces the variance of SGD by estimating differences in the SGD iterates, can improve the stochastic first order (SFO) oracle complexity for this problem by improving the previously known bound of $O((n + n^{2/3} \kappa_x \kappa_y^2)\log\frac1\epsilon)$ to $O((n + n^{1/2} \kappa_x \kappa_y^2)\log\frac1\epsilon)$. Further improvements are provided for ill-conditioned instances.

**Questions:**

Minor comments

- Defintion 3.1: should say $y\in\mathbb R^{d_y}$
- Assumption 3.1: is there an extra $L^2$ in the parentheses?
- Line 427 (supplementary): typo “us a question thta whether”

**Limitations:**

This is largely a theoretical paper and has no potential for negative societal impact.

**Strengths And Weaknesses:**

The obvious strength of this work is that it provides a direct improvement in the SFO oracle complexity for the smooth minimax optimization problem under PL conditions by improving previous results by poly(n) factors, which is significant. Because I do not work in the optimization literature, I could not tell what technical innovations were necessary to make this result possible, and it was not clear to me from the discussion of the paper either. In particular, the only conceptual message I got from the work was that “SPIDER-type algorithms can improve algorithms for PL minimax optimization beyond the previous SVRG-based algorithm.” This is certainly an important message, but I would appreciate more discussion and intuition on what this result implies in the broader theory of minimax optimization algorithms. For example, the authors point out previous works that apply SPIDER-type algorithms to minimax optimization, under different assumptions on the function being optimized, and mention that the technical details are different since earlier works converge sublinearly while the current work converges linearly; however, it is not clear to me how significant this difference is, since such differences already appear in basic analyses of vanilla gradient descent for convex vs strongly convex functions. In particular, SREDA in Luo et al. [23] achieve rates of the form $O(n\log(\kappa/\epsilon) + n^{1/2}\kappa^2 / \epsilon^2)$, which seems very similar to the guarantee of the current paper, and a further discussion of the differences would be appreciated.

Pros

- Direct improvements over previous work on an important problem in minimax optimization theory.
- Extremely detailed and clean presentation of proofs in the supplementary material.

Cons

- Lack of discussion which places the innovations in this work in the context of other work in minimax optimization.

---

> ### Author Response · Authors · 2022-07-31
> **Reply to Reviewer vsqY**
>
> We thank the reviewer's effort and helpful comments.
>
> We clarify the difference between SREDA [23] and the results in Section 6 of our paper.
>
> 1. Luo et al. [23] suppose the objective function is strongly-convex in $y$ and possibly nonconvex in $x$, while the algorithms in Section 6 of our paper only suppose the objective function satisfies PL condition in $y$, which is weaker than strong concavity.
>
> 2. Section 6 of our paper conduct the Catalyst acceleration to reduce the dependency on condition number, which is not considered by Luo et al. [23]. As a result, the SFO complexity shown in Theorem 6.2 of our paper only has $\kappa\log(\kappa)$ dependency on condition number, which is better than $\kappa^2$ dependency obtained by SREDA.
>
> 3. The implementation of our SPIDER-GDA is much easier than SREDA.
> Note that SPIDER-GDA iterates with $x$ and $y$ simultaneously.
> However, SREDA requires a concave maximizer to iterate on $y$ after per update on $x$, which leads to an additional inner loop that makes the implementation be complicated.
>
> Thank you for pointing out the typos. We have fixed them in revision.

---

> > ### Comment · Reviewer_vsqY · 2022-08-07
> > **Thank you for the responses.**
> >
> > Thank you for explaining the differences between SREDA and the current paper.
> >
> > I found the authors' response to Reviewer NEh5's comments to be particular insightful in some of the technical contributions of this work, rather than the current response, and I think it is an interesting contribution.

---

> > > ### Author Response · Authors · 2022-08-08
> > > **Reply**
> > >
> > > Thanks for your comment! We are glad that Reviewer vsqY have read our response to Reviewer NEh5 and found our contribution is interesting. We would like to incorporate these discussions into our later version.

---

### Official Review · Reviewer_N7QK · 2022-07-07

**Rating:** 6
**Confidence:** 2
**Soundness:** 3 good
**Presentation:** 3 good
**Contribution:** 3 good

**Summary:**

The paper studies minimax optimization under Polyak-Łojasiewicz (PL) conditions. Given a function $f(x, y) = \frac{1}{n}\sum_{i=1}^{n}f_i(x, y)$ that is $\mu_x$-PL in $x$ and $\mu_y$-PL in $y$, assuming that each $f_i$ is $L$-smooth, the algorithm provably finds an $\epsilon$-approximate solution for $\min_x\max_y f(x, y)$ using $O((n + \sqrt{n}\kappa_x\kappa^2_y)\log(1/\epsilon))$ queries to a stochastic first-order oracle, where $\kappa_x = L/\mu_x$ and $\kappa_y = L/\mu_y$ are the condition numbers. Prior bounds have a larger factor of $n^{2/3}$ instead of $\sqrt{n}$ in front of the condition number. The authors also present an accelerated algorithm, which achieves a better query complexity of $\tilde O(\kappa_x\kappa_y\sqrt{n}\log^2(1/\epsilon))$ in the $\kappa_x \ge \sqrt{n}$ case.

The first algorithm, termed "SPIDER-GDA", is a variance-reduced version of alternating gradient descent ascent. The second algorithm proceeds by using SPIDER-GDA as a basic solver, and applying Catalyst acceleration to it.

In addition, the authors discuss an extension of the techniques to the setting where the function only satisfies the PL condition in $y$ but not $x$. Experiments are performed on synthetic data (quadratic functions), and the proposed algorithms are shown to converge faster than a baseline of Yang et al. (2020).

**Questions:**

I have a few clarifying questions regarding the novelty and significance of the work, and I might update the rating based on the answer:

1. What are the main differences between SPIDER-GDA and the SVRG-AGDA algorithm of Yang et al. (2020)? (E.g., is it about using a larger minibatch of size $B \approx \sqrt{n}$ instead of $B = 1$?)

2. Are these changes novel? E.g., are they applied in other settings of stochastic first-order optimization? And if so, does this work involve a novel/different analysis?

3. To what extent are these changes necessary? E.g., is there evidence proving/suggesting that the $n^{2/3}$ dependence is unavoidable for SVRG-AGDA?

**Limitations:**

The main limitation is that the optimality of the proposed algorithms remains unclear. This is stated by the authors in Section 8.

**Strengths And Weaknesses:**

Strengths: The paper is well written. I think the authors did a good job in motivating the problem, rigorously defining the problem setting, and presenting the results. Compared to the prior work of Yang et al. (2020), the query complexity of the new algorithms improves by a polynomial factor in theory, and is also shown to outperforming empirically. Overall, this is a solid and well-presented work.

Weaknesses: On the negative side, the current paper is lacking in novelty and significance. Considering that: (1) the techniques used in the paper (variance reduction, alternating gradient descent ascent, Catalyst acceleration) are all well-known in the optimization literature; and (2) the query complexity obtained could still be far from optimal, I doubt whether the present work meets the bar.

The presentation could be further improved if the authors give more intuition/explanation behind the improvement from $n^{2/3}$ to $\sqrt{n}$; see questions below.

In addition, there are a couple of possible math typos:
- Page 1, Line 3: The optimization problem differs from (1) by a factor of $n$.
- Page 1, Line 10: Should $\log(1/\epsilon)$ be $\log^2(1/\epsilon)$ instead?
- Page 1, Line 10: It might be better to clarify what $\tilde O$ notation hides: polylog factors in $\kappa_x$, $\kappa_y$ but not $1/\epsilon$?
- Page 4, Line 91: Extra $L^2$ in parentheses?
- Page 7, Table 3: Missing ")" for SPIDER-GDA, first case

---

> ### Author Response · Authors · 2022-07-31
> **Reply to Reviewer N7QK**
>
> We thank the reviewer's effort and helpful comments.
>
> 1. There are two-main difference between SPIDER-GDA and SVRG-AGDA.
>
> $\quad$ a). The gradient estimators of SPIDER-GDA and SVRG-AGDA are different.
> We omit the subscript $t$ in remain.
> For SPIDER-GDA, the gradient estimator at $k$-th round depends on the nearest $(k-1)$-th round (see line 10-11 of Algorithm 1).
> For SVRG-GDA, the gradient estimator at $k$-th round depends on the gradient at $0$-th round for each $k$ (line 9-10 of Algorithm 5 in Appendix C). Intuitively, the point $(x_k,y_k)$ could be too far away from $(x_0,y_0)$ for large $k$, which leads to the estimation error of SVRG-AGDA be larger than SPIDER-GDA. As a result, SPIDER-GDA has lower complexity than SVRG-AGDA.
>
> $\quad$ b). The update schemes of SPIDER-GDA and SVRG-AGDA are different.
> SVRG-AGDA use the alternating update, that is
> $x_{k+1} = x_k - \eta_x G_x(x_k,y_k)$ and $y_{k+1} = y_k + \eta_y G_y(x_{k+1},y_k)$.
> In contrast, SPIDER-GDA use simultaneous update, that is $x_{k+1} = x_k - \eta_x G_x(x_k,x_k)$ and $y_{k+1} = y_k + \eta_y G_y(x_k,y_k)$.
> The convergence analysis of simultaneous-type algorithm SPIDER-GDA is much
> simpler than SVRG-AGDA.
> Note that proof of Theorem D.1 in our paper is much simpler than the proof of similar result for AGDA-SVRG (page 26-28 in arXiv:2002.09621, which is the full version of [42]).
> Additionally, we also provide GDA-SVRG in Appendix C (the simultaneous-type algorithm with SVRG estimator), which has the same order of complexity as AGDA-SVRG [42]. Similarly, the analysis of GDA-SVRG in our framework is much simpler than AGDA-SVRG.
>
> 2. The techniques of our paper are quite different with other setting of stochastic first-order optimization.
> Compared with minimization problem, the minimax problem studied in our paper is more difficult since we have to consider two variables. Concretely, the analysis of proposed SPIDER-GDA target to show $A_k = g(x_k) - g(x^*)$ and $B_k = f(x_k,y_k) - g(x_k)$ converge to zero simultaneously.
> We also requires the gradient estimators well approximates $(\nabla g(x_k), - \nabla_y f(x_k,y_k))^\top$, rather than only showing they approximate $(\nabla_x f(x_k,y_k), -\nabla_y f(x_k,y_k))^\top$, where $g(x) = \max_y f(x,y)$.
> The difference between $g(x_k)$ and $f(x_k,y_k)$ makes our theoretical analysis more challenging.
> Another related problem is solving nonconvex-strongly-concave minimax problem by stochastic first-order algorithms. Please see the response to ``Reviewer vsqY'' for the discussion.
>
> 3. Our paper presents SPIDER-GDA by using mini-batch size $\mathcal{O}(\sqrt{n})$ and the stepsizes $\tau_x=\mathcal{O}(1/ (\kappa_y^2 L))$, $\tau_y=\mathcal{O}(1/L)$.
> The same SFO upper complexity of SPIDER-GDA also can be obtained by using mini-batch size $\mathcal{O}(1)$ and stepsizes  $\tau_x=\mathcal{O}(1/(\sqrt{n} \kappa_y^2 L))$, $\tau_y=\mathcal{O}(1/(\sqrt{n} L))$. General speaking, using larger stepsizes could reduce the number of iteration, but each of iteration requires the larger mini-batch size.
> The total number of SFO oracle calls will not be changed by such adjustment if we balance the parameter of batch-size and stepsizes appropriately.
> The main reason of SPIDER-GDA can improve the factor from $n^{2/3}$ to $\sqrt{n}$ is owing to the different type of gradient tracking, which is discussed in 1.
>
> 4. For nonconvex minimization, a plenty of SVRG-type algorithms have $n^{2/3}$ factor in their SFO upper bound complexity.
> Hence, we think it is reasonable that SVRG-AGDA also needs $n^{2/3}$  dependency.
> To the best of our knowledge, there is no rigorous theory to show whether $n^{2/3}$ dependence is unavoidable for SVRG-type algorithms (even for minimization, there is no such theory). We think this is an interesting problem in feature direction.
>
> 5. Thank you for pointing out the typos. We have fixed them in revision.

---

> > ### Comment · Area_Chair_Lfpa · 2022-08-08
> > **Please acknowledge the authors' reply**
> >
> > Please acknowledge the authors' reply.

---

> > ### Comment · Reviewer_N7QK · 2022-08-08
> > **Thank you for your reply!**
> >
> > Thank you for the clarification! I thought they helped me better understand the difference between the new algorithm and the previous ones, as well as the challenge posed by the minimax optimization (compared to minimization).
> >
> > Given the response, I feel that the contribution of the current work is fairly solid, and have (temporarily) updated the overall score to 6 (weak accept) accordingly. I don't have further questions for the authors.

---

> > > ### Author Response · Authors · 2022-08-09
> > > **Thanks for your reply!**
> > >
> > > Thank Reviewer N7QK for the time and effort. We are glad that the reviewer appreciated our clarification.

---

### Official Review · Reviewer_NEh5 · 2022-07-10

**Rating:** 5
**Confidence:** 3
**Soundness:** 3 good
**Presentation:** 3 good
**Contribution:** 2 fair

**Summary:**

The paper is devoted to non-convex stochastic (finite sum) saddle point problems under PL assumption (two-side and one-side). The paper proposes a new method based on the famous variance reduction SPIDER. A modification with acceleration via Catalyst is also given.

**Questions:**

1) Did the authors try to consider loopless variants of the SVRG (L - SVRG) and SPIDER (PAGE) methods? For minimization problems the proofs for them are simpler in my opinion. Perhaps it would be the same here. If such variants have not been considered, it could be explored as future works.

2) For example, SVRG methods have a direct accelerated version - Katyusha or L-Katyusha. Did the authors try to design methods with direct acceleration, without envelopes? If not, it could be explored as future works.

**Limitations:**

No limitations and potential negative societal impact

**Strengths And Weaknesses:**

**Strengths:**

1) Solving non-convex stochastic problems is an important task. The paper is relevant and interesting.

2) It is easy for me to follow the paper. The literary review is done on a quite good level.

3) The paper improves on existing bounds in the literature.

4) In my opinion, the way to get the result is not trivial. But perhaps if one use L-SVRG or PAGE (loopless modifications of SVRG and PAGE) as a base it would be easier.

**Weaknesses:**

1) Literature review: if we are talking about SPIDER it has a lot of analogues in the literature -- see Table 1 from

Li, Z., Bao, H., Zhang, X. Richtarik, P. PAGE: A Simple and Optimal Probabilistic Gradient Estimator for Nonconvex Optimization.

I definitely ask for a citation of the next work, because it seems that SPIDER is just a version of SARAH:

Nguyen, L. M., Liu, J., Scheinberg, K., and Takac, M. SARAH: A novel method for machine learning problems using stochastic recursive gradient.

2) The result of the paper is expected if one knows what estimates look like for minimization problems under PL conditions:
GD --- SVRG --- SPIDER: $\mathcal{O}\left( n \cdot L/\mu\right)$ --- $\mathcal{O}\left( n^{2/3} \cdot L/\mu\right)$ --- $\mathcal{O}\left( \sqrt{n} \cdot L/\mu\right)$.
For saddle point problems we observe the same dependencies on $n$ -- GD and SVRG the results were already known.

3) The PL condition does not seem to be particularly popular and widely researched (for saddle point problems), and its applicability to practical problems is slightly blurred (despite the fact that the authors give references to some works)

**Verdict before discussions:**

The work makes an improvement on the available results for an important task but in not the most popular assumptions. The results are expected and understandable. I wouldn't be opposed to this work being presented at the conference, but I wouldn't be upset if it not. Borderline accept

---

> ### Author Response · Authors · 2022-07-31
> **Reply to Reviewer NEh5**
>
> We thank the reviewer's effort and helpful comments.
>
> 1. We thank the review so much for pointing out the valuable references of PAGE and SARAH. We are happy to cite these papers in revision.
> Our work has not considered the loopless modification,
> but we strongly agree that such variant has potential to simplify the analysis and implementation.
> We are willing to explore this point as future works.
>
> 2. The existing results $\mathcal O(n\cdot L/\mu)$ and $\mathcal O(n^{2/3}\cdot L/\mu)$ of saddle point problems are obtained by alternating-type algorithms AGDA and SVRG-AGDA. These methods depend on the update rules of
> $x_{t+1} = x_t - \eta_x G_x(x_t,y_t)$ and $y_{t+1} = y_t + \eta_t G_y(x_{t+1},y_t)$, where $G_x$ and $G_y$ are the gradient (or gradient estimators) of $x$ and $y$ respectively. In contrast, our paper focus on the simultaneous gradient descent ascent, that is $x_{t+1} = x_t - \eta_x G_x(x_t,x_t)$ and $y_{t+1} = y_t + \eta_y G_y(x_t,y_t)$.
> The convergence analysis of simultaneous-type algorithm is so
> concise and its framework is quite different from previous work [42].
> As a comparison, we provide $O(n^{2/3}\cdot L/\mu)$ complexity for GDA-SVRG (simultaneous-type algorithm with SVRG estimator) in Appendix C.
> The proof of Theorem C.1 in our paper is much simpler than the proof of similar result for AGDA-SVRG (page 26-28 in arXiv:2002.09621, which is the full version of [42]).
> Similarly, the convergence analysis of proposed SPIDER-GDA with $\mathcal O(\sqrt{n}\cdot L/\mu)$ complexity is also concise.
> Although it is possible to design SARAH/SPIDER-type algorithm for AGDA to obtain $\mathcal O(\sqrt{n}\cdot L/\mu)$ complexity, the convergence analysis may be so complicated.
>
> 3. One of the popular applications of minimax problem with PL condition is AUC maximization within overparameterized neural network [13, 22]. Its objective function satisfies two-sided PL condition and has no convexity. Liu et al. [22] provided justifications for PL condition of AUC model with one hidden layer neural network in Appendix A.7 of their paper. The discussion of PL condition for more general case of deep AUC model can be found in Section 4 of  https://arxiv.org/pdf/2006.06889.pdf.
>
> 4. Direct acceleration without envelopes looks non-trivial in our setting. We agree that it could be explored as future works.

---

> > ### Comment · Reviewer_NEh5 · 2022-08-05
> > **Reply**
> >
> > Thanks to the authors for their response!
> >
> > 2) I understand that, which is why I find your results not trivial.
> >
> > I have no further questions for the authors. If they too have nothing to add, I am ready to move on to a discussion with the other reviewers and then make my final decision.

---

> > > ### Author Response · Authors · 2022-08-08
> > > **Reply**
> > >
> > > Thanks for your reply!  We are glad that Reviewer NEh5 found our results non-trivial.

---

### Official Review · Reviewer_dMKi · 2022-07-11

**Rating:** 5
**Confidence:** 1
**Soundness:** 3 good
**Presentation:** 3 good
**Contribution:** 2 fair

**Summary:**

This paper studies finite-sum minimax optimization problems under PL conditions. The SPIDER-GDA algorithm, which uses simultaneous Gradient Descent-Ascent as the backbone and SPIDER as its variance reduced gradient estimator, is proposed and analyzed, enjoying better guarantees than SVRG-based approaches. Catalyst acceleration and the one-sided PL extensions are also discussed.

**Questions:**

Is it possible to provide gradient complexity lower bounds for this setting?

**Limitations:**

Limitations of this work are adequately discussed.

**Strengths And Weaknesses:**

Strengths:
- The complexity guarantees improve the current state-of-the-art for both the two-sided and one-sided settings.
- The algorithms are fairly easy to understand and implement. The two-sided and one-sided settings are handled with the same algorithms with different hyper-parameters.

Weaknesses:
- The setting for this paper is somewhat niche, considering the finite-sum case with PL conditions and no convexity. What are some good examples or justifications for this setting? (The numerical experiments in Sec. 7, for instance, are quadratic and convex-concave).

---

> ### Author Response · Authors · 2022-07-31
> **Reply to Reviewer dMKi**
>
> We thank the reviewer's effort and helpful comments.
>
> 1. The model of AUC maximization within overparameterized neural network [13, 22] is a good sample. Its objective function satisfies two-sided PL condition and has no convexity. Liu et al. [22] provided justifications for PL condition of AUC model with one hidden layer neural network in Appendix A.7 of their paper. The discussion of PL condition for more general case of deep AUC model can be found in Section 4 of  https://arxiv.org/pdf/2006.06889.pdf.
>
> 2. The gradient lower bounds for minimax optimization with PL condition is still an open problem. To the best of our knowledge, even for the minimization problem, the optimality of first-order methods under PL condition is unclear. We believe this is an interesting topic in future.

---

> > ### Comment · Area_Chair_Lfpa · 2022-08-08
> > **Please acknowledge the authors' reply**
> >
> > Please acknowledge the authors' reply.

---

> > ### Comment · Reviewer_dMKi · 2022-08-08
> > **Reply**
> >
> > Thanks to authors for providing the example of overparameterized networks and for the explanation on gradient lower bounds. I have no further questions for the authors.
> >
> > I still feel that the contribution of this paper is somewhat limited by the restricted applicability of the setting and will maintain my score.

---

### Meta-Review · Area_Chair_Lfpa · 2022-08-23

**Recommendation:** Accept
**Confidence:** Certain

**Metareview:**

This paper present an algorithm with strong theoretical guarantees for a fundamental problem of broad interest.  It is well-written.


**Award:**

No

---

### Decision · Program_Chairs · 2022-09-14

Accept